# Poly ADP-ribosylation of SET8 leads to aberrant H4K20 methylation in mammalian nuclear genome

Pierre-Olivier Estève[1,3], Sagnik Sen [1,3], Udayakumar S. Vishnu[1], Cristian Ruse[1,2], Hang Gyeong Chin[1] & Sriharsa Pradhan [1✉]

In mammalian cells, SET8 mediated Histone H4 Lys 20 monomethylation (H4K20me1) has been implicated in regulating mitotic condensation, DNA replication, DNA damage response, and gene expression. Here we show SET8, the only known enzyme for H4K20me1 is post-translationally poly ADP-ribosylated by PARP1 on lysine residues. PARP1 interacts with SET8 in a cell cycle-dependent manner. Poly ADP-ribosylation on SET8 renders it catalytically compromised, and degradation via ubiquitylation pathway. Knockdown of PARP1 led to an increase of SET8 protein levels, leading to aberrant H4K20me1 and H4K20me3 domains in the genome. H4K20me1 is associated with higher gene transcription levels while the increase of H4K20me3 levels was predominant in DNA repeat elements. Hence, SET8 mediated chromatin remodeling in mammalian cells are modulated by poly ADP-ribosylation by PARP1.

---

[1] New England Biolabs Inc, 240 County Road, Ipswich, MA 01938, USA. [2] Present address: Moderna Therapeutics, 200 Technology Square, Cambridge, MA 02139, USA. [3] These authors contributed equally: Pierre-Olivier Estève, Sagnik Sen. ✉email: pradhan@neb.com

In the eukaryotic nucleus, double-stranded DNA is wrapped around histone octamers to make chromatin fibers. These chromatin fibers undergo various structural changes during cell division and organism development[1–3]. The decompaction and re-establishment of chromatin organization immediately after mitosis are essential for genome regulation[4]. Given the absence of membranes separating intranuclear substructures and during cell division, it has been postulated that other structural features of the nucleus, such as the chromatin itself, must impart regulations that would control molecular information flow and stage different physiological activities. Indeed, chromatin accessibility plays a central role in ensuring cell cycle exit and the terminal differentiation during metamorphosis[5]. Furthermore, chromatin compaction threshold in cells exiting mitosis ensures genome integrity by limiting replication licensing in G1 phase. Recent studies on histone writer enzymes have shade some lights on the role of histone marks in cell biology[6]. The nucleosome structure illustrates that highly basic histone amino (N)-terminal tails can protrude from the core and would be in direct contact with adjacent nucleosomes. Therefore, modification of these histone tails would affect inter-nucleosomal interactions, and thus would modulate the overall chromatin 3D-structure and compaction. Recent advances in Hi-C studies support that this is indeed the case, where transcriptionally active A and inactive B compartments are differentially positioned in the nucleus[7]. Histone modifications not only regulate chromatin structure by merely being there, but they also recruit chromatin readers and remodeling enzymes that utilizes the energy derived from the hydrolysis of ATP to reposition nucleosomes[6]. For example, lysine acetylation of histone tails is dynamic and regulated by the opposing action of two families of enzymes, histone acetyltransferases (HATs) for acetylation, and histone deacetylases (HDACs) to remove the acetyl groups[8]. Indeed, acetylated histones on the chromatin are a cue to transcriptional gene activation[9]. Therefore, the balancing act between both HATs and HDACs is essential in maintaining the dynamic equilibrium during gene expression.

The other significant chromatin mark, histone methylation, primarily occurs on the side chains of lysines and arginines[10]. Histone methylation does not alter the charge of the histone protein, unlike phosphorylation. However, the epsilon amino group of lysine may be modified to mono-, di- or tri-methylated configuration, whereas arginine may be mono-, symmetrically or asymmetrically di-methylated, creating an unprecedented array of complexity for the reader proteins. These core modifications are established and propagated by SET domain containing enzymes[11]. One such enzyme is SET8 (also known as KMT5A, PR-Set7, and SETD8), the only histone methyltransferase that monomethylates histone H4K20 (H4K20me1)[12,13]. Subsequent modification of H4K20me1 by Suv4-20h1/h2 leads to the transition from H4K20me1 to H4K20me2/3[14]. Although H4K20-specific demethylases are less understood, PHF8 is the only known enzyme that demethylates H4K20me1 to H4K20[15]. Similarly, Rad23 has also been shown to demethylate H4K20me1/2/3[16]. In the mouse genome, H4K20 methylation state is distributed in specific regions and is bound by specific reader proteins. Histone H4K20me1 has been implicated in regulating diverse processes ranging from the DNA damage response, mitotic condensation, DNA replication, and gene regulation. Indeed, loss of the SET8 causes a more severe and complex phenotype, as this negates the catalysis of all levels of H4K20 methylation. Loss of SET8 causes lethality at the third instar larval stage in Drosophila[17], and in mice it leads to embryonic lethality via developmental arrest between the four- and eight-cell stages[12]. Rescue experiments on mouse SET8-/- phenotype embryos by reintroduction of either catalytically active or inactive allele have demonstrated that the catalytic activity of the enzyme is essential for embryo development.

SET8 protein expression is tightly regulated during the cell cycle; it is highest during G2/M and early G1 and is absent during the S phase. Indeed, upon mitotic exit, chromatin relaxation is controlled by SET8-dependent methylation of histone H4K20. In the absence of either SET8 or H4K20me1, substantial genome-wide chromatin decompaction occurs allowing excessive loading of the origin recognition complex (ORC) in the daughter cells[18]. During cell cycle, SET8 undergoes ubiquitination and phosphorylation. SET8 is a direct substrate of the E3 ubiquitin ligase complex CRL4Cdt2[19–21]. In addition, SET8 is also regulated by the E3 ubiquitin ligase SCF/Skp2[22]. Skp2 degrades substrates during S and G2 phase and may partially contribute to the steep decrease in SET8 levels during the S phase[23]. The other post-translational modification of SET8 that may affect its activity or stability is serine phosphorylation, as discovered by proteomics studies[24–26]. SET8 S29 is phosphorylated in vivo by Cyclin B/cdk1 during mitosis, and it causes SET8 to dissociate from mitotic chromosomes at anaphase and relocate to the extrachromosomal space, where it is dephosphorylated by cdc14 phosphatase and subsequently subjected to ubiquitination by APC/Cdh1[27]. Therefore, the coordination of ubiquitylation and phosphorylation processes may be necessary to maintain precise levels of H4K20me1 and SET8 in the genome.

Apart from acetylation, methylation, phosphorylation, and ubiquitylation, poly ADP-ribosylation is another post-translational modification of proteins catalyzed by PARP family of enzymes by the addition of linear or branched chains of ADP-ribose units, from NAD+ substrate[28]. The central enzyme for poly ADP-ribosylation in cells during DNA damage is poly-ADP-ribose polymerase 1 (PARP1). Multiple different amino acids are shown to be acceptors of PAR, such as Lys, Arg, Glu, Asp, Cys, Ser, Thr, pSer (phospho-serine, through the phosphate group), although His and Tyr residues were also proposed by proteomic approaches[29–35]. PARP family members modify many proteins by poly and/or mono ADP-ribosylation[36]. PARPs catalyze short or long, branched, or linear ribosylation. The structural diversity of poly ADP-ribosylation on these proteins could be recognized by poly ADP-ribosylation readers (or binders) with their specific binding motifs for biological function.

In a proteomics study of SET8 binding proteins, we found PARP1 as a strong binder (Supplementary Table 1). This led us to study the nature of SET8-PARP1 interaction and the role of poly ADP-ribosylation on the catalytic activity of enzymes. Here, we have also investigated the role of poly ADP-ribosylation in SET8 degradation, chromatin remodeling, and aberrant H4K20 methylation in mammalian cells.

## Results

**PARP1 ribosylates SET8**. In a proteomic analysis of SET8 pull-down in HEK293T cells, we discovered that PARP1 is a strong binder (Supplementary Table 1). To reconfirm this observation, we performed reciprocal immunoprecipitation either with anti-SET8 or anti-PARP1 antibody, performed western blots, and probed with respective antibodies. Indeed, PARP1 pulled down SET8 and vice-versa, compared to IgG control (Fig. 1a). This led us to investigate if this interaction is cell cycle-dependent, since SET8 expression is regulated by cell cycle[37]. For this experiment, we transfected COS-7 cells with FLAG-PARP1 and GFP-SET8, synchronized the cells, and studied their association using confocal microscopy and Pearson's correlation coefficient during G1, S, and G2/M stages. At G1, GFP-SET8 and FLAG-PARP1 remain distributed throughout the nucleus, compared to S phase where punctate pattern of both GFP-SET8 and FLAG-PARP1 was

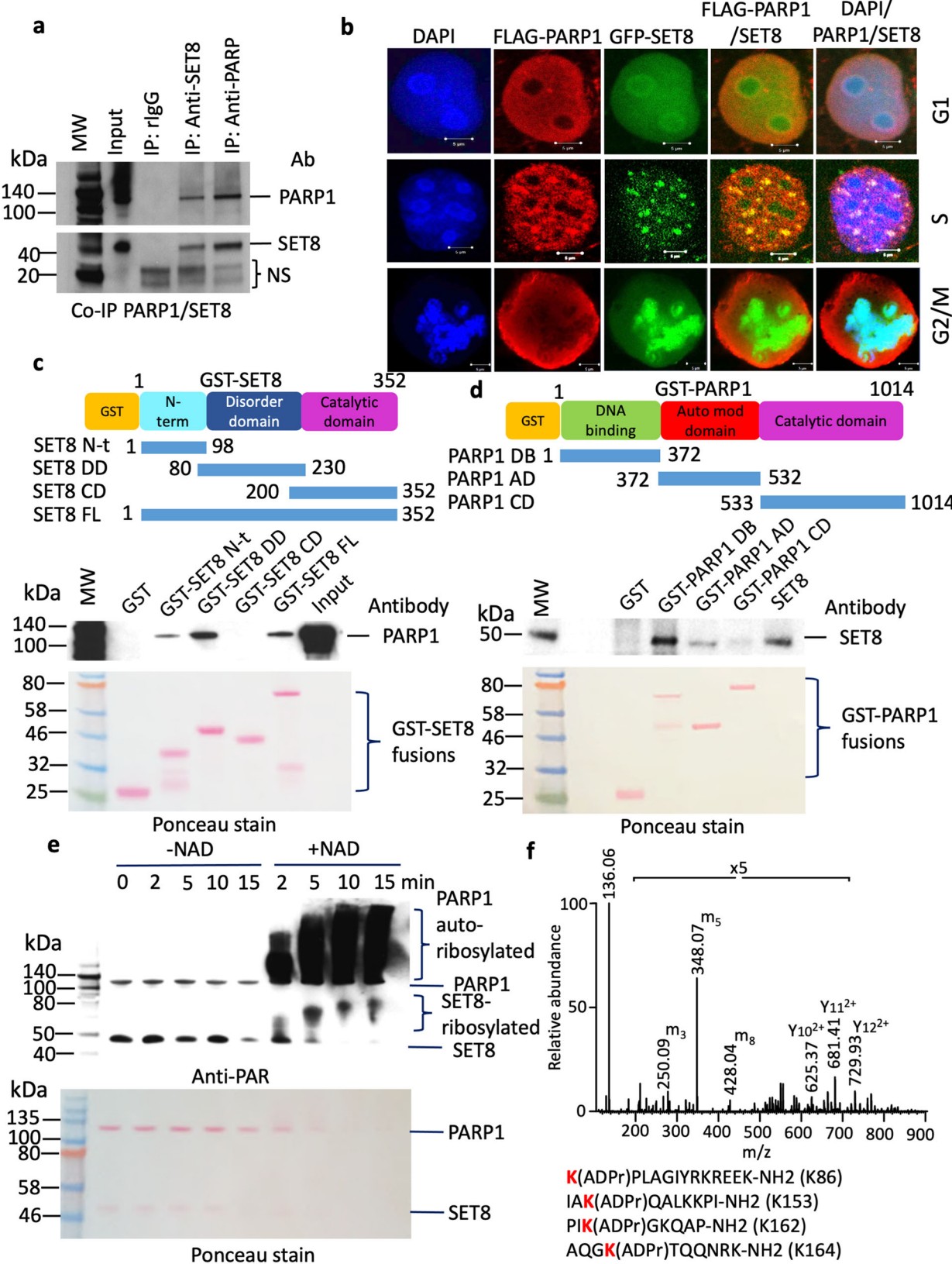

observed. However, as expected FLAG-PARP1 remained throughout the nucleus, and appeared as prominent punctate foci with GFP-SET8, as observed by bright yellow merged spots (Fig. 1b). This was further supported by their association kinetics during cell cycle where the Pearson's correlation was higher during S phase ($r = 0.6$) compared to G1 ($r = 0.3$) and no correlation was observed in G2/M phase (Supplementary Fig. 1a, Supplementary Data 4). The same cells were also probed with an anti-PCNA antibody to examine the chromatin replication foci. Indeed, PCNA remained punctate in the nucleus, so as GFP-SET8, and both were colocalized at the onset of S phase (Supplementary Fig. 1b). These observations suggest that PARP1 has

**Fig. 1 Colocalization, binding, and poly ADP-ribosylation of SET8 by PARP1. a** Co-immunoprecipitation of endogenous SET8 and PARP1 in HCT116 cells. Rabbit IgG (left lane) was used as a negative control. Non-specific bands are represented as NS. **b** Colocalization between FLAG-PARP1 (red) and GFP-SET8 (green) during cell cycle in COS-7 cells. DAPI (blue) represents the nuclear DNA content. **c, d** Mapping of domain interactions using GST-pulldown assays between SET8 domains (top, left) and full-length recombinant PARP1 protein and GST-pulldown assays between PARP1 domains and full-length recombinant SET8 protein (top, right). The PARP1 or SET8 binding were detected by western blotting using PARP1 antibody (middle, left) or SET8 antibody (middle, right), respectively. Ponceau stain gels (bottom) represent the amount of GST beads constructs used for the GST-pulldown assays. **e** In vitro detection of full-length recombinant SET8 ADP-ribosylation by full-length recombinant PARP1 by western blotting using anti-ADP ribose antibody (top). Ponceau stain gels (bottom) represent the amount of PARP1 and SET8 recombinant enzyme used for ADP-ribosylation assay (bottom). **f** Detection of SET8 lysines ADP-ribosylation using mass spectrometry analysis of SET8 ADP-ribosylated peptides by full-length recombinant PARP1 protein in vitro.

multiple targets throughout the nucleus, but also has strong colocalization with SET8 at the chromatin replication foci. To narrow down the exact binding motifs between PARP1 and SET8, we performed a reciprocal GST-pulldown assay. Immobilized GST fusion of overlapping SET8 fragments covering the entire protein was challenged with full-length PARP1. After stringent washes, the bound proteins were denatured in SDS gel loading buffer and separated on SDS-PAGE followed by blotting onto a membrane for western blot with anti-PARP1 antibody to determine the binding domain. Two fragments covering both N-terminal 1–98 amino acids and disorder domain 80–230 amino acids residues showed binding. Although, a stronger binding was observed with the disorder domain (DD) (Fig. 1c). A similar reciprocal GST pulldown experiment using immobilized GST-PARP1 fusion fragments covering the entire protein was challenged with full-length SET8. The bound proteins were western blotted and probed for SET8 to reveal GST-PARP1 DNA binding domain (GST-PARP1 DB) as the strongest binder (Fig. 1d). Based on the results of both the colocalization and GST pulldown experiments, we concluded that both SET8 and PARP1 are indeed binding partners.

We next investigated the functional consequence of this interaction. Both purified SET8 and PARP1 were incubated together in the absence or presence of cofactor NAD+ for various lengths of time, spanning between 2 and 15 min. The reactions were stopped, and the proteins were separated on SDS PAGE, western blotted and probed with anti-PAR (anti-poly ADP ribose) antibody. If the proteins are ribosylated, an anti-PAR antibody will show higher molecular weight shift for the proteins. We observed auto-poly-ADP-ribosylation of PARP1 as expected, only in the presence of NAD+ (Fig. 1e; Supplementary Fig. 2a). We also observed a strong reaction time-dependent SET8 poly ADP-ribosylation. Indeed, all SET8 molecules were poly ADP-ribosylated by PARP1 within 10 min of reaction as observed by high molecular weight migrating smear (Fig. 1e). We further investigated the poly ADP-ribosylated regions of SET8 by performing in vitro poly ADP-ribosylation assay of overlapping peptides followed by western blotting and probing the blot with anti-PAR antibody. We found that GST-SET8 DD is indeed the substrate of PARP1 (Supplementary Fig. 2b). We narrowed down the amino acid residues 81–98 and 157–180 as the putative acceptor of ADP-ribose for poly ADP-ribosylation (Supplementary Fig. 2c, d).

**LC-MS analysis of PARP1 activity on SET8 peptides**. To determine the nature of poly-ADP-ribosylation and the acceptor amino acid, we made synthetic peptides covering 86–98 and 158–170 amino acids and performed in vitro poly ADP-ribosylation, and analyzed the reaction product using LC-MS. First, we monitored the PARP1 activity with SET8 peptide KPLAGIYRKREEK-NH2 (86–98 aa). Peptides with an amidated C-terminus require specific conditions for detection to support MS/MS sequencing identification[38]. We detected and quantified signals for the three main isotopes starting with 0 min (negative control) and

incubation with PARP1 for 1 h, 4 h, and overnight (Supplementary Fig. 3a). We observed a decrease in the signal of the KPLAGIYRKREEK-NH2 peptide after the initial 1 h incubation. Concomitantly, we assessed the peak areas for two charged states (+3, +4) of the (ADPr) KPLAGIYRKREEK-NH2 in order to detect the activity of PARP1 (Supplementary Fig. 3b, c). Three main isotopes for each charge state showed the signal of (ADPr) KPLAGIYRKREEK-NH2 present at 1 hr followed by a decrease in intensity at 4 h and overnight reactions. This suggested the formation of additional species beyond the initial addition of one ADP-ribosylation unit. Next, we investigated the peptide signals at two faster PARP1 reaction times, 5 min and 15 min. Chromatographic ion signals for (1ADPr)KPLAGIYRKREEK-NH2 were detected for both charge states +3 and +4 at retention time 8.35 min (Supplementary Fig. 3d). The minimal chromatographic signal was detected for (2ADPr)KPLAGIYRKREEK-NH2 with an increased elution time of 10.33 min. At 15 min reaction time, the signal for di-ADP-ribosylated peptide near 10.3 min increased with a wider base (Supplementary Fig. 3e). A second peak with intermediate hydrophobicity relative to the mono ADP-ribosylated peptide showed the most intense di-ADP-ribosylation signal at RT 9.77 min (Supplementary Fig. 3e). Therefore, we detected the formation of possible diastereomers of increased hydrophobicity upon addition of 1ADPr unit to mono ADP-ribosylated peptides during a 10 min reaction time interval.

Characterization of MS/MS HCD fragmentation pattern of (ADPr) KPLAGIYRKREEK-NH2 peptide is presented in Supplementary Fig. 3. For charge state +3, we observed the formation of fragment ions m3, m5, and m8 that are independent of the peptide sequence and indicative of the ADP-ribose group (Supplementary Fig. 4a, top panel). The nomenclature for ADP ribosylation fragment ions is according to Hengel et al.[39]. Similarly, but with lower intensity, charge state +4 showed confirmed the presence of these fragment ions m3, m5, and m8 (Supplementary Fig. 4b, bottom panel). Intact peptide backbone ions containing partial ADP ribose group and peptide y sequencing ions can be visualized at $m/z$ 500–900 (Supplementary Fig. 3a, b, bottom panels; Supplementary Fig. 3b is the same spectrum from Fig. 1f). In addition to HCD spectra, CID spectra can be probed for ADP-ribosylation information of amidated C-terminus peptides. We screened seven amino acids for the position of the ADPr unit (Supplementary Fig. 5). For both charges +3 and +4, the associated MS/MS spectra mapped with a higher score the N-terminal lysine of KPLAGIYRKREEK-NH2.

Analysis of SET8 peptide KKPIKGKQAPRKK-NH2 (158–170 aa) showed that activity of PARP1 resulted in maximum signal for 1ADPr unit addition at charge +4 for overnight reaction (Supplementary Fig. 6a). Detection of lower intensity peak areas for all three main isotopes of charge +4 together with moderate signal decrease of unmodified KKPIKGKQAPRKK-NH2 (Supplementary Fig. 6b) suggested a slower PARP1 kinetic towards this SET8 peptide. For comparison, CID MS/MS spectrum of this charge state with unit resolution linear ion trap showed the presence of ADP-ribose signature ions (Supplementary Fig. 7). Taken together,

we found multiple Lys residues were poly ADP-ribosylated, including but not limited to K86, K158, K162, and K164 (Fig. 1f).

**Mutation of lysine residues in SET8 alters the disorder domain.** SET8 has a prominent disordered domain with PONDR-VLXT score above 0.5 and fold index lower than 0 (Supplementary Fig. 8a, b). The disorder domain binds strongly to PARP1 (Fig. 1c). It is composed of charged amino acids residues such as lysines that are also poly ADP ribosylation accepter (Fig. 1f). To determine the role of the amino acids and poly ADP-ribosylation in PARP1 binding, GST-SET8 fusion was mutated at K86A, K158/159A, K162/164A, R168A, K169/170A, K174A (GST-SET8 M). The disorder domain of the SET8 M displayed ordered structure with PONDR score below 0.5 suggesting the mutant lysine residues facilitating this event (Supplementary Fig. 8a, c vs. b, d). Furthermore, a brief network algorithm-guided structure space analysis was performed to determine the effect of the structural modification due to amino acids substitution. Structures of the monomeric SET8 and SET8 M were modeled using I-TASSER prediction tool (Supplementary Fig. 9). The modeled structure displayed a prominent binding cleft in the wild type compared to the mutant (Supplementary Fig. 9a, b). Overall, the distance among the original Lys/Arg or substituted Ala residues at positions 86, 158, 159, 162, 164, 168, 169, 170, and 174 (within the domain of SET8) was lower for SET8 M (Supplementary Fig. 9c–f; Supplementary Table 2). Subsequently, normal mode-based GNM utilized the residual oscillation scores for studying the structural modification. These scores were also used to determine the weight between two amino acids residues (given as nodes) which can further be considered as co-oscillation possibility. The edge betweenness-based clustering model considered these weights for shortest path calculation. Therefore, the centrality score assigned for each residue could be used as structural dependency on those residues. In Supplementary Fig. 10, significant modifications of betweenness centrality score were observed (Supplementary Fig. 10a, b right panels). As per the distributions of the scores throughout the structures, localize residue-specific dependencies were higher in SET8 M comparing to wild type which could be elaborated more through module detection. The number of modules were 10 and 12 for SET8 and SET8 M, respectively (Supplementary Fig. 11). The incremental number of modules can be considered as the indication of minor orderedness where the member of each clusters helped to analyze it further. Cluster 5 from wild-type enzyme had all the lysine-enriched residues from disordered regions. However, these residues are distributed in three different clusters, 5, 8, and 9 in SET8m. Thus, it is plausible that SET8M has higher propensity of localized orderedness than wild-type enzyme that may affect protein–protein interaction and other macromolecule binding. To test our hypothesis, we performed GST-pulldown assay by incubating purified PARP1 with the full-length GST fusion SET8 (GST-SET8 FL) or the mutant (GST-SET8 M). After several washes the bound proteins were separated on SDS-PAGE, western blotted and probed with anti-PARP1 antibody. We observed significant (~70%) loss of PARP1 binding, confirming SET8 DD lysine residues are indeed essential for this interaction (Supplementary Fig. 12, Supplementary Data 6).

**PARP1 Poly ADP-ribosylates SET8 impacting DNA and nucleosome binding.** SET8 is well known for its interaction with nuclear proteins, PCNA, a processivity factor involved in DNA replication and required for S-phase progression[37,40]. A crystallography study demonstrated that SET8 employs its i-SET and c-SET domains to engage nucleosomal DNA 1 to 1.5 turns from the nucleosomal dyad[41]. To evaluate the DNA binding activity of SET8, we incubated a 100 bp DNA ladder with various GST-fusion fragments of SET8, GST-SET8 FL, and GST-SET8 M. After the incubation time, we resolved fusion proteins-DNA or nucleosome complexes on the gel. If a defined size of DNA is bound with protein, that DNA band will be shifted and will not be represented on the gel compared to the control ladder. As predicted by crystallography, GST-SET8 FL protein bound predominately to double-stranded DNA ranging between 100 and 300 bp in the gel-shift assay. This binding was partially dependent on the 157–352 amino acids of SET8, and a significant loss of binding was observed in a deletion mutant comprising 175–352 amino acids, suggesting 157–175 amino acids play a functional role in DNA binding (Fig. 2a, Supplementary Fig. 13a). We also performed gel shift assay of purified SET8 protein with a fluorescent-labeled double-stranded DNA oligo and measured the Kd values of 1.6 +/− 0.6 μM (Fig. 2b). After observing the robust DNA binding activity of SET8, we also validated its binding activity with recombinant mono-nucleosomes using similar GST-SET8 fusions that were used for DNA binding activity studies shown in Fig. 2a. Once again, 157–175 amino acids played a functional role in nucleosome binding (Fig. 2c, Supplementary Fig. 13b). Taken together, data from previous crystallography studies and our gel-shift assays, we conclusively demonstrated that SET8 has both double-stranded DNA and nucleosome binding activity.

Since SET8 is a substrate for PARP1, we investigated the role of DNA in SET8-PARP1 binary complex formation, and whether poly ADP-ribosylation can affect either DNA or nucleosome binding activity of SET8. DNA is a catalytic activator of PARP1, we incubated GST-SET8, DNA, and PARP1 in the presence and absence of DNase I and performed GST-pulldown assay followed by western blotting and probing the bound proteins with respective antibodies. Indeed, both GST-SET8 and PARP1 formed binary complexes in the presence of double-stranded DNA. However, in the presence of DNase I, SET8-PARP1 binding was reduced confirming DNA as a facilitator of binary complex formation (Supplementary Fig. 14a, b, Supplementary Data 6).

**PARP1 poly ADP-ribosylates SET8 impacting catalytic activity.** Since SET8 mutant displayed ~70% loss of PARP1 binding compared to the wild type of enzyme and the mutations were in poly ADP-ribosylation acceptor amino acids, we hypothesized that ADP-ribosylation residues would impact its substrate binding, thus impacting catalytic activity. To examine our hypothesis, we incubated SET8 with either a 100 bp ladder or with a recombinant mononucleosome in the presence of PARP1 and NAD+ for poly ADP-ribosylation, and the control remained without NAD+ cofactor. Indeed, poly ADP-ribosylated SET8 lost the DNA as well as nucleosome binding activity as observed by prominent DNA or nucleosome bands (Fig. 2d, top panel; Supplementary Fig. 15a, b). To confirm that this observation is due to poly ADP-ribosylation, a portion of the same reaction was western blotted and probed with anti-PAR antibody. Indeed, the SET8 poly ADP-ribosylation was evident in those lanes that had poor binding of DNA or nucleosome with GST-SET8 (Fig. 2d middle panel).

Next, we evaluated if SET8 methyltransferase activity on histone H4 is modulated by PARP1-mediated poly ADP-ribosylation of the enzyme. We performed in vitro histone methyltransferase assays mimicking either poly ADP-ribosylated SET8 or its unmodified form. To initiate poly ADP-ribosylation of SET8, we pre-incubated SET8, PARP1, DNA, and NAD+ for 15 min at room temperature, followed by the addition of methyl donor, tritiated AdoMet, and purified recombinant histone H4 to

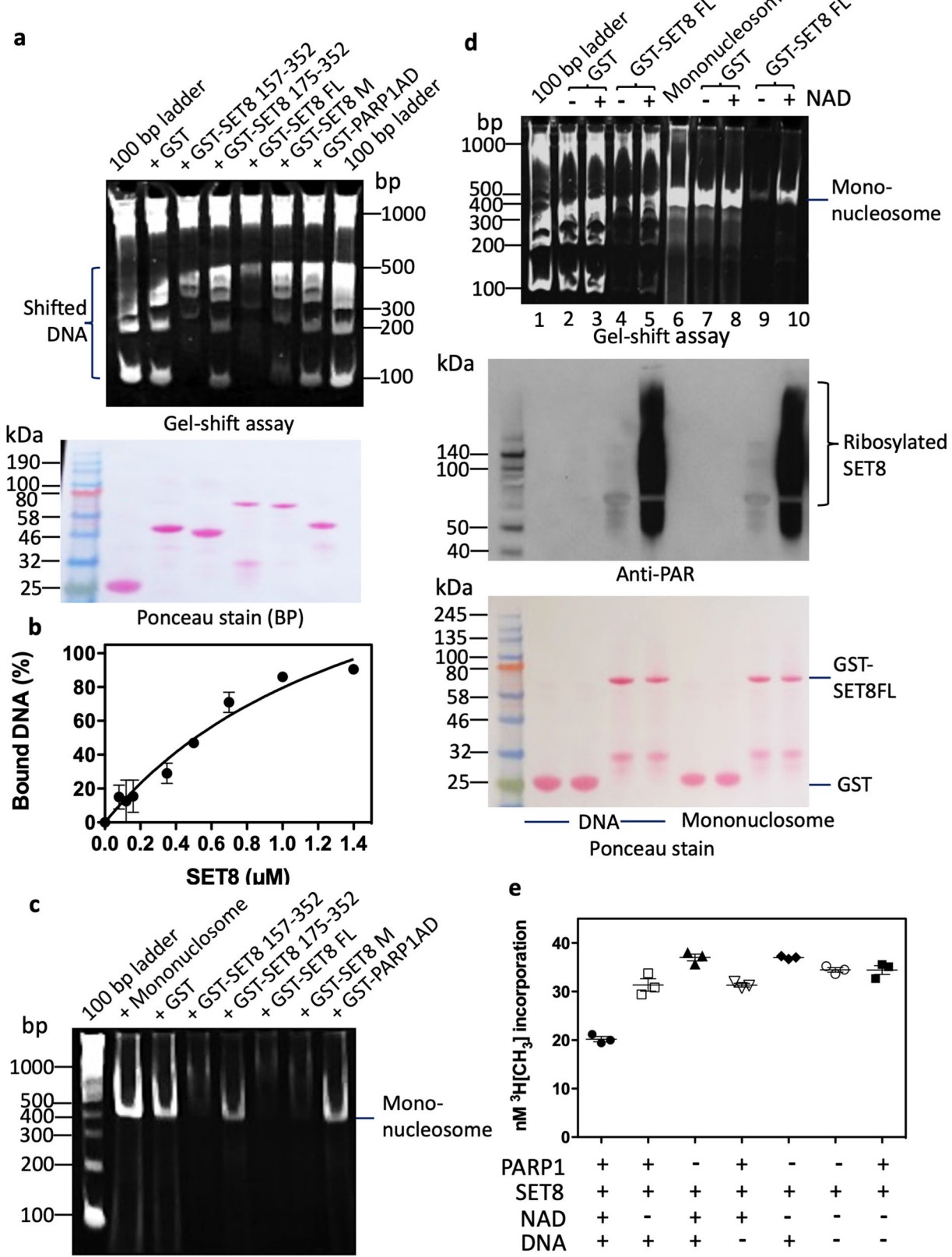

initiate H4K20 monomethylation. After the reaction, we performed the filter binding assay for tritium incorporation on substrate histone H4. Indeed, poly ADP-ribosylation SET8 lost ~40–50% activity compared to the control that lacked any one component for successful poly ADP-ribosylation (Fig. 2e). These results suggest that PARP1 not only inhibits SET8 binding to DNA and nucleosome, but it can also impact histone H4K20 monomethylation.

**Poly ADP-ribosylation promotes SET8 degradation.** Poly ADP-ribosylation was recently identified as a signal for triggering protein

**Fig. 2 Poly ADP-ribosylation impairs DNA, nucleosome binding, and catalytic activity of SET8 protein. a** Detection of unbound 100 bp DNA ladder by TBE ethidium bromide-stained gel in supernatants (left lane) on GST-SET8 domains or mutant (M) using GST-pulldown assays (top, left side). Asterisks are representing shift of DNA on GST-SET8 157–352 amino acid protein or GST-SET8 full-length protein (FL) beads. Ponceau stain represents the amount of GST beads constructs used for the GST-pulldown (bottom, left side). **b** Different concentrations of recombinant full-length SET8 protein binding to DNA using EMSA to determine the equilibrium dissociation constant ($K_d$). **c** Detection of unbound mononucleosome by TBE ethidium bromide-stained gel in supernatants on GST beads versus GST-SET8 domains or mutant (M) beads using GST-pulldown assays. **d** Detection of unbound DNA (left side) or unbound mononucleosome (right side) by TBE ethidium bromide-stained gel in supernatants (top) on GST beads versus GST-SET8 FL beads using GST-pulldown assays. GST or GST-SET8 FL beads were poly ADP-ribosylated or not (with or without NAD) using full-length recombinant PARP1 as demonstrated by western blot using anti-ADP ribose antibody (middle). Ponceau stain gel (bottom) represents the amount of GST bead constructs used for the GST-pulldown and the ADP-ribosylation western blot analysis. **e** SET8 histone methyltransferase assay on full-length recombinant histone H4 using full-length recombinant SET8 in the presence or absence of activated full-length recombinant PARP1.

degradation through the ubiquitin-proteasome system. Indeed, poly ADP-ribosylation-mediated degradation of ARTD1 (ADP-ribosyltransferase diphtheria toxin-like 1) is documented[42]. In another example, PARP1 binds and poly ADP-ribosylates bromodomain-containing protein 7 (BRD7), which enhances its ubiquitination and degradation through the PAR-binding E3 ubiquitin ligase RNF146[43]. This led us to investigate if PARP1 has any modulating effect on SET8 levels in the cell. First, we transfected cells with either FLAG or FLAG-PARP1 construct and measured the transfection efficiency by western blotting and probing with anti-FLAG antibody, followed by probing the blot with anti-H4K20me1, 2, and 3 along with SET8. Indeed, PARP1 overexpression had reduced SET8 and its reaction products, H4K20me1 to almost half of the control (Supplementary Fig. 16a, b, Supplementary Data 7). We then systematically transfected HA-ubiquitin with GFP-SET8 or FLAG-PARP1 alone or co-transfected all three constructs into the mammalian cells and treated the cells with proteasome inhibitor MG132. We performed western blots of cell extracts to evaluate transfection efficiency and expression of constructs. Probing the blot with anti-GFP antibody demonstrated equivalent expression GFP-SET8 fusion in all transfected samples. Similarly, anti-FLAG antibody probing of the blot showed robust expression of FLAG-PARP1 fusion protein (Fig. 3a, upper panel). Next, we immunoprecipitated GFP-SET8 fusion with anti-GFP antibody and probed the western blot with anti-HA to detect HA-ubiquitinated SET8 fusion or anti-PAR for poly ADP-ribosylated SET8 fusions. Indeed, GFP-SET8 showed high molecular weight HA-ubiquitinated smear when expressed alone with HA-ubiquitin (Fig. 3a, lower left panel, lane 1). GFP-SET8 co-expression with FLAG-PARP1 and HA-ubiquitin resulted in accumulation of high molecular weight HA-ubiquitinated smear, suggesting that the expression of additional PARP1 enhances ubiquitination of GFP-SET8 (Fig. 3a, lower left panel, lane 2, 1.6x). In addition, reducing the degradation of ubiquitin-conjugated GFP-SET8 by MG132 resulted in higher amounts of high molecular weight HA-ubiquitinated smear as expected. When we co-expressed GST-SET8, FLAG-PARP1, and HA-ubiquitin and reduced protein degradation using MG132, we observed ~5 folds accumulation of HA-ubiquitinated GST-SET8 protein (Fig. 3a, lower left panel, lane 4, 2.3x). Similarly, ribosylated GFP-SET8 was more prominent in GFP-SET8 and FLAG-PARP1 overexpressed cells (Fig. 3a, lower right panel, lane 2 and 4, 2.4x and 3.6x, respectively). All these experiments conclusively prove that PARP1 is an effector protein that aids in poly ADP-ribosylation of SET8 leading to ubiquitin-mediated degradation.

The SET8 is ubiquitinated on chromatin by CRL4(Cdt2) complexes during S phase and following DNA damage in a PCNA-dependent manner. In a transgenic mouse model for lung cancer, the level of SET8 was reduced in the preneoplastic and adenocarcinomous lesions following overexpression of Cul4A[44]. Therefore, it is expected that downregulation of Cul4A by siRNA would lead to accumulation of SET8. Similarly, downregulation of

PARG, that removes ADP ribose from poly ADP-ribosylated protein by PARP1 and acts as an antagonist of PARP1, would also lead to the accumulation of poly-ribosylated SET8. To validate our hypothesis, we transfected GFP-SET8 fusion constructs to mammalian cells and treated the cells with DMSO, Cul4A inhibitor (MLN4924), or PARG inhibitor (PDD00017273) and monitored the SET8 pattern of expression by immunoprecipitation and western blotting. Indeed, GFP-SET8 fusion protein was expressed in a similar level in the absence or presence of either Cul4A or PARG inhibitor (Fig. 3b, top panel). Immunoprecipitation of GFP-SET8 showed more prominent high molecular weight poly ADP-ribosylated GFP-SET8 fusion in PARGi or Cul4Ai -treated cells, 1.6x and 2.5x, respectively (Fig. 3b, 3rd panel). When these samples were western blotted and probed with anti-HA antibody to decipher ubiquitin abundance, we observed almost 40% reduction in ubiquitination confirming Cul4A inhibitor (MLN4924) essentially act by inhibiting ubiquitination pathway. Similarly, upon PARGi treatment 1.6x GFP-SET8 accumulation is visible by high molecular weight poly-ubiquitinated fusion enzyme (Fig. 3b, 4th panel).

To confirm the natural occurrence of poly ADP-ribosylation indeed occurs in endogenous SET8 enzyme during cell growth, we treated HeLa cells with MG132 to reduce protein degradation and immunoprecipitated SET8 with anti-SET8 antibody. The control antibody was anti-GFP. The immune precipitates were separated on SDS-PAGE, western blotted and probed with anti-ADPribose and anti-SET8. Immunoprecipitated samples displayed a strong high molecular weight smear of SET8, although MG132 treated sample had higher intensity confirming poly-ADP ribosylation is crucial for SET8 degradation (Fig. 3c, left panel, Supplementary Data 3). Quantitative measurements of relative band intensities showed ~40% more signal intensity in the presence of MG132 for SET8 (Fig. 3c, right panel). Taken together with results from Fig. 3a–c, we correlate that high molecular weight SET8 or its GFP fusion are both ubiquitinylated and poly ADP-ribosylated, suggesting a cross-communication between both post translational modifications to maintain a steady-state level of SET8 in the cells.

**Poly ADP-ribosylation and ubiquitinylation regulate SET8 levels and H4K20 methylation.** Since there was a correlation between poly ADP-ribosylation and ubiquitination of SET8 molecules, we transfected both GFP-SET8 FL or GFP-SET8 M into HeLa cells in the presence of HA-ubiquitin plasmid and monitored levels of fusion SET8, ADP-ribosylated fusion SET8 and ubiquitinylated-SET8. Indeed, wild-type fusion displayed prominent high molecular weight ladder pattern indicating poly ADP-ribosylated fusion SET8, correlating with ubiquitinated-SET8 and the levels were higher ~3.6x compared to the mutant enzyme, since the acceptor Lys residues were mutated (Fig. 4a). This led us to hypothesize there is a synergistic effect of poly ADP-ribosylation and poly ubiquitination in SET8 stability. We

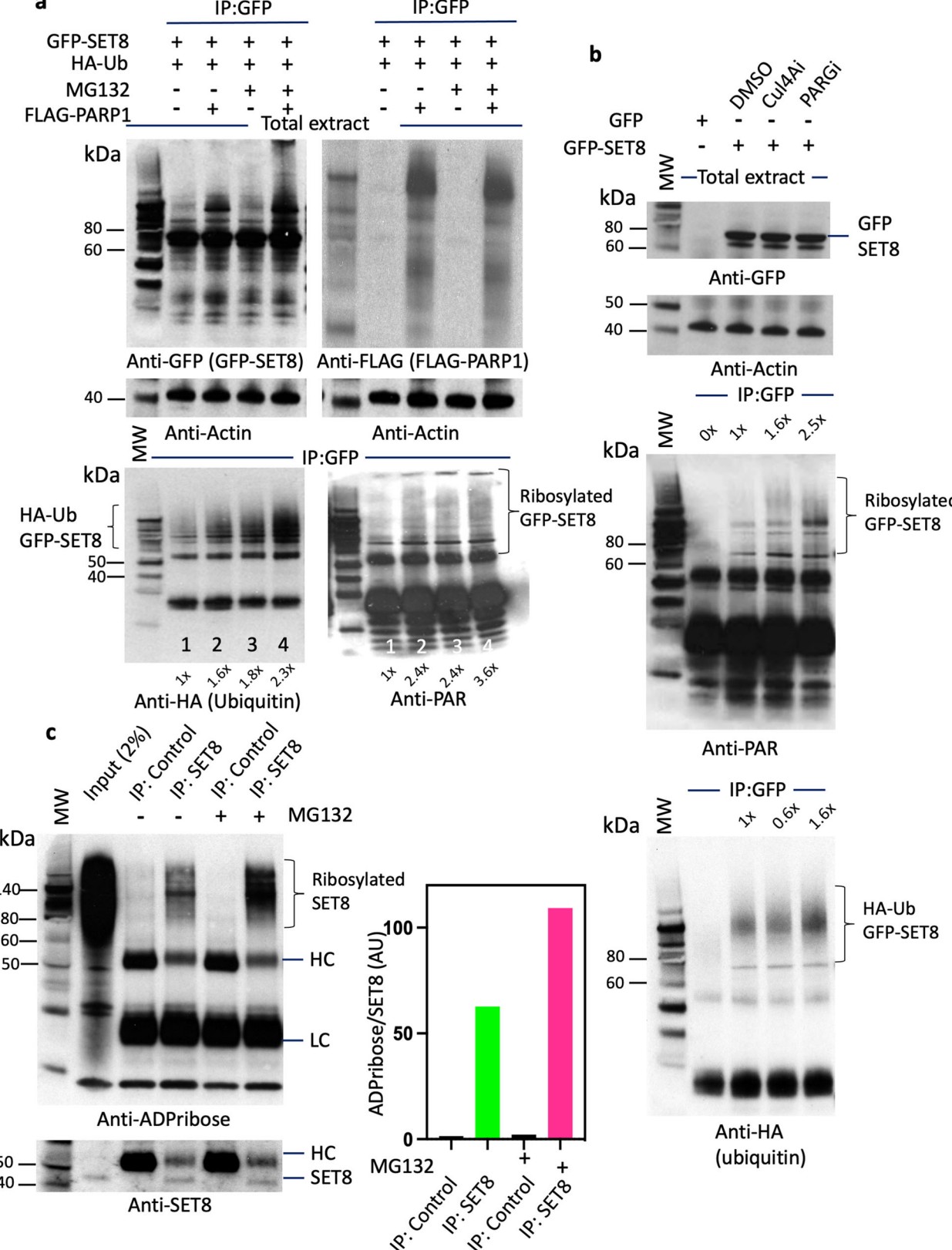

performed half-life measurement for both GFP-SET8 FL and GFP-SET8 M in the presence of cycloheximide. Indeed, wild-type SET8 fusion degraded much faster compared to the mutant enzyme, although the loading control, actin remain constant throughout the time course. The wild-type GFP-SET8 had a half-life of 3.8 h compared to 12.3 h for the mutant GFP-SET8

(Fig. 4b). Therefore, it is plausible that the lysine residues in the disordered domain are crucial in maintaining SET8 enzyme levels in cell. We next pursued half-life studies of endogenous SET8 by PARP1 depletion using siPARP1. Half-life of endogenous SET8 in control cells with siGFP transfection was 0.74 h compared to 1.47 h in PARP1 knockdown cells (Supplementary Fig. 17a, b,

**Fig. 3 PARP1 promotes SET8 protein degradation. a** GFP-SET8 immunoprecipitation in overexpressed GFP-SET8 COS-7 cells with or without FLAG-PARP1 overexpression in presence or not of the proteasome inhibitor MG132. Western blots detecting the amount of GFP-SET8 protein overexpressed in total extract (top, left) as well as the amount of FLAG-PARP1 protein overexpressed (top, right) using anti-GFP and anti-FLAG antibody, respectively. Western blots detecting the amount of ubiquitin (Ub) (bottom, left) or ADP-ribosylation (bottom, right) of immunoprecipitated GFP-SET8 protein using anti-HA and anti-ADP ribose antibody, respectively. Anti-actin was used as a loading control (middle). The fold increase was calculated by densitometry and indicated at the bottom of the western blot. **b** GFP-SET8 immunoprecipitation in overexpressed GFP-SET8 COS-7 cells in presence or not (DMSO) of Cullin inhibitor (Cul4Ai) or PARG inhibitor (PARGi). Western blot detection of total GFP-SET8 protein levels overexpressed in COS-7 cells (1st panel). Anti-actin antibody was used as control (2nd panel). Western blot detecting the amount of poly-ADP ribosylation, using anti-PAR antibody (3rd panel) or the amounts of HA-ubiquitin using anti-HA antibody (4th panel) of immunoprecipitated GFP-SET8 protein. **c** Endogenous SET8 immunoprecipitation in HeLa cells in presence or absence of the proteasome inhibitor MG132. Western blot (left side) detecting the amount of ADP-ribosylation in GFP (IP: control), SET8 immunoprecipitates (top panel) as well as the amount of SET8 protein (bottom panel). Respective densitometry analyses of SET8 ADP-ribosylation abundance representative of at least 2 biological experiments are shown (right side; $n = 2$).

Supplementary Data 5). The half-life of endogenous SET8 of 0.74 h matched with previously reported study[19]. The discrepancy of half-life between GFP-SET8 (3.8 h) vs. endogenous SET8 (0.74 h) could be due to the stability of GFP fusion partner of SET8.

We next knocked down PARP1 using siRNA to define its role in SET8 stability and H4K20 modifications. HeLa cells were treated with control siGFP, siPARP1, and siSET8, and the extracts were western blotted and probed with respective antibodies. As expected, PARP1 siRNA was able to knock down 70% PARP1 resulting in 1.5-fold increase in SET8 protein level but not SET8 mRNA level confirming that SET8 increase was due to post translational modification (Supplementary Data 9). Surprisingly, SET8 level increase did not translate into an increase in global H4K20me1 level (Fig. 4c, Supplementary Data 8). This led us to investigate if global H4K20me2/3 heterochromatic marks are affected in the knockdown cells using western blot. Surprisingly, H4K20me2 level decreased along with the concurrent gain of global H4K20me3 suggesting knockdown of PARP1 facilitates the rapid formation of H4K20me3 that may result in aberrant heterochromatic mark establishment following cell cycle.

**Cell cycle-dependent interaction of SET8 and PARP1.** Since siPARP1 led to an increase in H4K20me3, and precursor H4K20me1 and SET8 enzyme are crucial to cell cycle progression genome stability, DNA replication, mitosis, and transcription, we hypothesized that SET8 and PARP1 would be colocalized in cells in a cell cycle-dependent manner for SET8 dynamics on chromatin. Lovastatin and thymidine-nocodazole-treated synchronized HeLa cell nuclei were isolated and extracts were made corresponding to G1, S, and G2/M phase. Equal amounts of extracts were evaluated for the relative abundance of SET8, PARP1, and H4K20me1 using western blot with respective antibodies. CDT1 (G1 phase marker) was used as a positive control for cell cycle synchronization. As expected, SET8 level was lowest during S phase and highest at G2/M phase, mirroring H4K20me1 levels. PARP1 level remained unchanged throughout the cell cycle (Fig. 5a, Supplementary Data 2). Since SET8 and PARP1 directly interact, we investigated SET8 levels on the chromatin. To validate this hypothesis, we used G1, S, G2/M nuclear extracts for SET8 immunoprecipitation (IP). The captured proteins were resolved on SDS-PAGE, western blotted and probed with SET8 and PARP1 antibodies (Fig. 5b, top panel). We observed the highest amounts of SET8 being captured in G2/M compared to S or G1 phase. SET8 IP also revealed that PARP1 is co-immunoprecipitated in G2/M compared to G1 or S phase. However, all three phases with PARP1 co-immunoprecipitated with SET8, abate a higher amount during S phase. We, therefore, quantified relative co-immunoprecipitation between PARP1-SET8 during all three cell cycle phases by comparing the ratios between PARP1 and SET8 in 3 independent experiments. Indeed,

PARP1/SET8 ratio was highest during S phase compared to the lowest in G2/M (Fig. 5b, below panel). Concurrently, in the same IP samples we also measured the ribosylated SET8 enzymes, the substrate of PARP1 (Fig. 5c). Indeed, the ratio of PARP1 and SET8 mirrored the ratio between ADP-ribosylated SET8 and SET8 confirming PARP1 modulates SET8 ribosylation which impacts the stability of the enzyme on the chromatin in a cell cycle-dependent manner (Fig. 5b vs. c). This would reflect SET8 enzyme level to global H4K20me1 during cell cycle.

To observe ADP ribosylation during cell cycle, we pulse changed COS-7 cells with 5-ethynyl-2'-deoxyuridine (EdU) to selectively labeled the newly synthesized DNA in S phase, for rapid visualization and its ability not to interfere with subsequent antibody staining. The same cells were also transfected with GFP-SET8 and further probed with anti-PAR antibody. EdU formed the punctate pink nuclear staining pattern, typical of S phase. And the same regions were also visualized as yellow indicating GFP-SET8 colocalization. Poly ADP-ribosylation was visualized using anti-PAR antibody as red throughout the nucleus, as expected with clear colocalization at the replication forks and SET8 (Fig. 5d). These results indicate poly ADP-ribosylation co-exists with SET8 on DNA replication forks and correlate with high level of SET8 ADP-ribosylation detected by IP (Fig. 5c). Thus, we conclude that PARP1 plays a central role during cell cycle to dynamically regulate H4K20 methylation.

**PARP1 level regulates H4K20me1 and H4K20me3 Chromatin Domains.** Since PARP1 predominantly interacts with SET8 during S phase leading to a concurrent decrease in SET8 and H4K20me1 level, we next investigated if there is a functional implication of PARP1 level in the cell and H4K20me1/me3 distribution on the chromatin. For this investigation, we knocked down PARP1 in HeLa cells and performed H4K20me1 and H4K20me3 ChIP-seq with respective siGFP controls. Prior to the ChIP-seq, we western blotted the samples and observed similar levels of SET8 and H4K20me1, me2, and me3 as observed for PARP1 knockdown (Fig. 4c). The Spearman correlation between the ChIP-seq fragments between control siGFP vs. knockdown siPARP was 0.95 and 0.95 for H4K20me1 and H4K20me3, respectively, demonstrating high degree of similarity. As expected, irrespective of siGFP or siPARP knockdown the correlation values between H4K20me1 and H4K20me3 remained below 0.37 suggesting both marks are mutually exclusive with a small percentage overlap. However, there was no significant similarity between H4K20me1 and me3 (Fig. 6a). We also mapped the H4K20me1 and me3 peaks and observed that PARP1 knockdown has higher read densities compared to their respective controls suggesting H4K20me1 hypermethylation in the peak regions and vicinity (−/+5 kbp), a similar pattern was observed for H4K20me3 hypermethylation (Fig. 6b, c). We performed peak annotations to determine the genomic elements associated with both post-translational marks. Indeed, LINE, SINE,

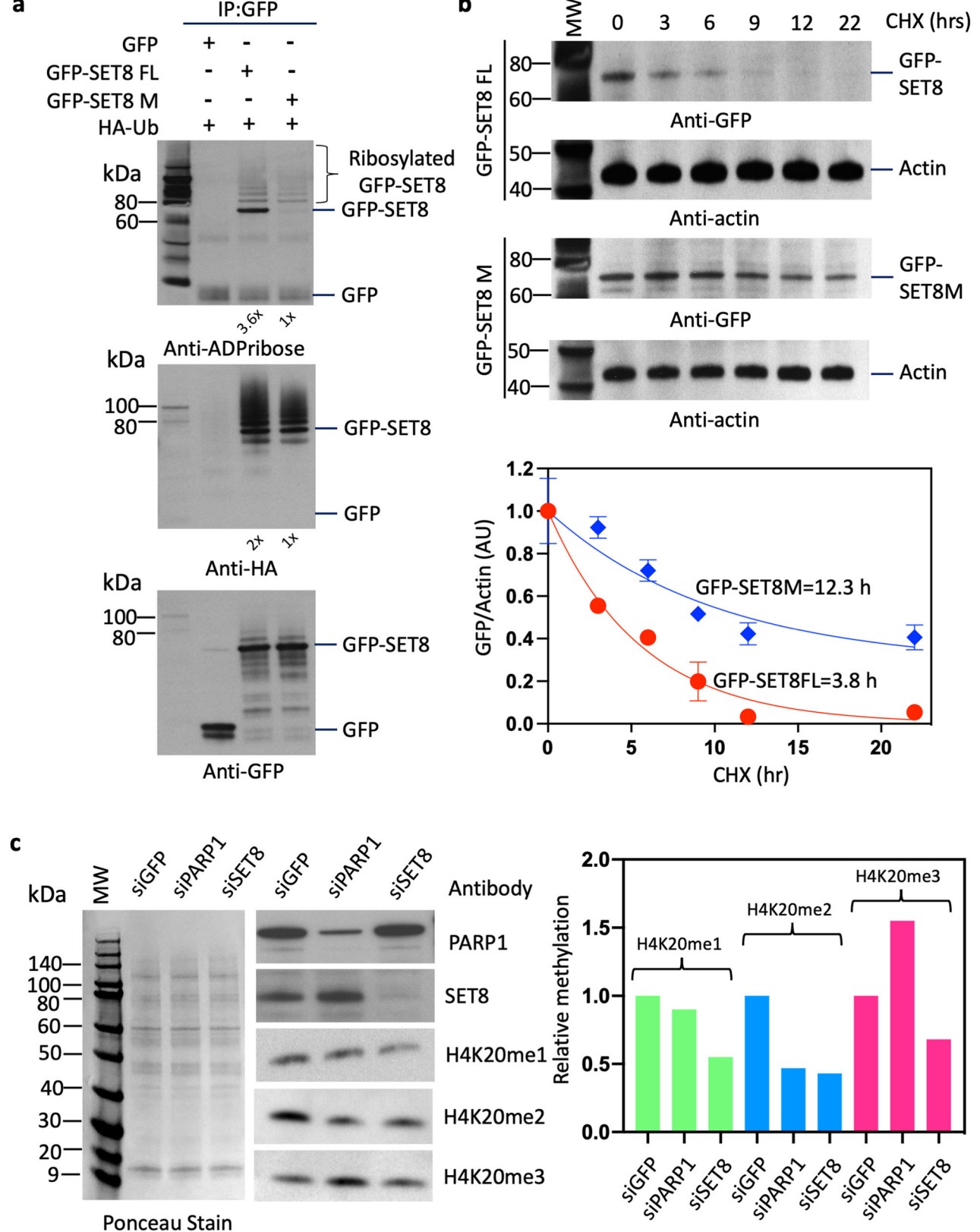

intron elements displayed appreciable amounts of hypermethylation for both H4K20me1 and me3. However, the satellites displayed loss of H4K20me1, and gain of H4K20me3 regional density in response to PARP1 knockdown, suggesting PARP1 levels have a different mechanism in maintaining the methylation status (Supplementary Fig. 18). The Spearman's correlation between the ChIP-

seq peaks between H4K20me1 and H4K20me3 for the above sets of genomic features remained below $r = 0.5$ for LINE, SINE, and intergenic regions. Surprisingly, the satellite regions had high degree of correlation indicating both H4K20me1 and me3 are enriched there as reported previously (Supplementary Fig. 19; refs. [45,46]).

**Fig. 4 PARP1 regulates SET8 protein stability. a** GFP-SET8 immunoprecipitation in overexpressed GFP-SET8 FL or GFP-SET8M with HA-Ubiquitin in COS-7 cells. Western blots detecting the amount of ADP-ribosylation (top panel), Ubiquitination (middle panel), and GFP fusion protein levels (bottom panel) in GFP immunoprecipitates. The fold increase was calculated by densitometry and indicated at the bottom of the western blot. **b** Cycloheximide chase analysis of SET8 stability in GFP-SET8 FL or GFP-SET8 M in HeLa cells. Western blots detecting the amount of GFP-SET8 FL and GFP-SET8 M protein levels with their respective actin levels (control) during cycloheximide time course (top panel). Respective densitometry analyses of GFP/Actin ratio representative of at least 2 biological experiments (bottom panel). **c** Western blots (left side) detecting the amount of PARP1 (top), SET8 protein (middle) as well as the amount of H4K20me1, H4K20me2, and H4K20me3 levels (bottom) in total protein extract of knockdown HeLa cells treated with siRNA (esiRNA) for GFP (control), esiRNA PARP1 and esiRNA SET8, respectively. Respective densitometry analyses of protein abundance representative of at least 2 biological experiments are shown (right side; $n = 2$). Ponceau stain was used as control (left side).

Further, we binned various peak width of H4K20me1 and H4K20me3 and observed that H4K20me1 distribution reduced below 1 kb peak width with concurrent gain up to 10 kb in response to PARP1 knockdown (Fig. 6d). Similarly, H4K20me3 distributions was more prominent on 500–1000 bp (Fig. 6e). This suggested that the knockdown of PARP1 may shift the dynamic equilibrium between both marks and thus changing the chromatin domains. Indeed, we observed more dramatic changes are in H4K20me1 domains. IGV browser displayed changes in H4K20me1 boundaries throughout the genome upon PARP1 knockdown (red segments) compared to control siGFP (blue segments) knockdown as observed for AKAP8 (Fig. 6f). To study the global gene expression corresponding to H4K20me1, ChIP-seq derived sequences annotated genes identified. The percentage overlap of the genes is shown based on the intersecting genes taken from respective RNA-seq and the intragenic regions of H4K20me1 annotated files. The intersecting genes in various conditions, siPARP1_K20me1, siGFP_K20me1, were 9.59%, 4.89%, respectively, confirming H4K20me1 has 2-folds more association with transcriptionally active genes (Supplementary Fig. 20a). Indeed, the logarithmic distribution of the positively expressed genes from ChIP-seq of H4K20me1 show more compatibility with global gene expression in siPARP1 cells as displayed in the density plots compared to control siGFP (Supplementary Fig. 20b).

## Discussion
The principal enzyme of poly ADP-ribosylation is PARP1. In this study, poly ADP-ribosylation not only impaired SET8 catalytic activity, it also ribosylated and facilitated SET8 degradation by the ubiquitin degradation pathway. As expected, knockdown of PARP1 resulted in accumulation of SET8 in cells, although it did not increase global H4K20me1, the enriched peaks displayed small hypermethylation at proximal and distal regions of the peaks. However, there was a profound decrease of global H4K20me2 and corresponding increase in global H4K20me3. This is not unexpected since H4K20me2 is the precursor of heterochromatin mark H4K20me3. In gene annotation analysis, the satellite DNA that comprises 10–15% of the human genome showed a decrease in H3K20me1 and corresponding increase of H4K20me3. This was further supported by the FLAG-PARP1 overexpression studies. Indeed, overexpression of PARP1 not only decreased the level of SET8, it also correspondingly decreased the level of H4K20me1, H4K20me2, and H4K20me3. A reasonable explanation would point that the knockdown cells were mostly in resting phase with saturated amounts of H4K20me1 on the genome. And secondly, the decrease in H4K20me2 and concurrent increase in H4K20me3 may be as a result of another methylase that is catalytically regulated by PARP1, which would need additional studies. Taken together, these results suggest an alteration in the level of SET8 in cells can profoundly impact heterochromatic mark H4K20me3.

An increase in H4K20me3 may have resulted in heterochromatic domain rearrangement in overexpressing cells. While many

studies have reported that PARP1 promotes gene transcription, it also promotes gene expression at the post-transcriptional level by modulating the RNA-binding protein HuR[47,48]. Indeed, PARP-1 is required for a series of molecular outcomes at the promoters of PARP-1-regulated genes, leading to a permissive chromatin environment for RNA Pol II machinery loading. Poly ADP-ribosylation has an important role in the maintenance of H3K4me3, as the enzyme for demethylation, KDM5B, is impaired by poly ADP-ribosylation. Consistently, an increased level of KDM5B at TSS of active genes is associated with decreased H3K4me3 after inhibition of poly ADP-ribosylation in vivo[49]. There is also in vitro evidence of a direct involvement of poly ADP-ribosylation in the crosstalk between H3 and H1 methylation. Indeed, poly ADP-ribosylation of H3 impairs its methylation by the H3K4 mono-methyltransferase SET7/9, thus shifting its catalytic activity towards other lysine residues of H1[50]. Another role of poly ADP-ribosylation can contribute to transcriptional repression by H3K9me2 accumulation at retinoic acid (RA)-dependent genes. In this mechanism, demethylase KDM4D is covalently poly ADP-ribosylation at the N-term domain, impairing its recruitment onto RA-responsive promoters, leading to repression by H3K9me2 accumulation[51]. Apart from the transcriptional role of PARP, inhibition/depletion of the enzymes also causes loss of epigenetic marker on heterochromatin, H3K9me3[52], H4K20me3[53], and 5mC[54] at the centromeric regions. These all studies suggest that PARP enzymes can modulate chromatin structure and regulate gene expression.

However, previous studies have reported poly ADP-ribosylated PARP1 might conflict with CG methylation by non-covalent interaction with DNMT1 preventing its access on DNA and catalysis[54]. PARP1 also affects DNA methylation by forming a complex with the transcription factor CTCF. In another report, PARP1 was shown to control the UHRF1-mediated ubiquitination of DNMT1 to timely regulate its abundance during S and G2 phase, thus impacting CG methylation. Therefore, the above observations, and our current studies, demonstrate other epigenetic marks particularly DNA methylation and SET8 levels in the cell, may play a role in H4K20me1 and H4K20me3 chromatin domains. Indeed, SET8 is known to regulate chromatin compaction during G1 phase[18]. Therefore, it would make sense that during this phase there is little interaction between PARP1 and SET8. Indeed, upon mitotic exit, chromatin relaxation is controlled by SET8-dependent methylation of histone H4K20. In the absence of either SET8 or H4K20me1 mark, substantial genome-wide chromatin decompaction occurs allowing excessive loading of the origin recognition complex (ORC) in the daughter cells[18]. Based on these results, it is plausible that PARP1 overexpression in cancer would catalytically compromise SET8 leading to slight reduction in H4K20me1 mark. This would have a larger ramification in an increase of H4K20me3 level on heterochromatin. Since H4K20me3 represses transcription when present at promoters and silences repetitive DNA and transposons, its aberrant deposition on chromatin may alter the transcriptional network during oncogenesis. Comparison of H4K20me1 ChIP-seq with

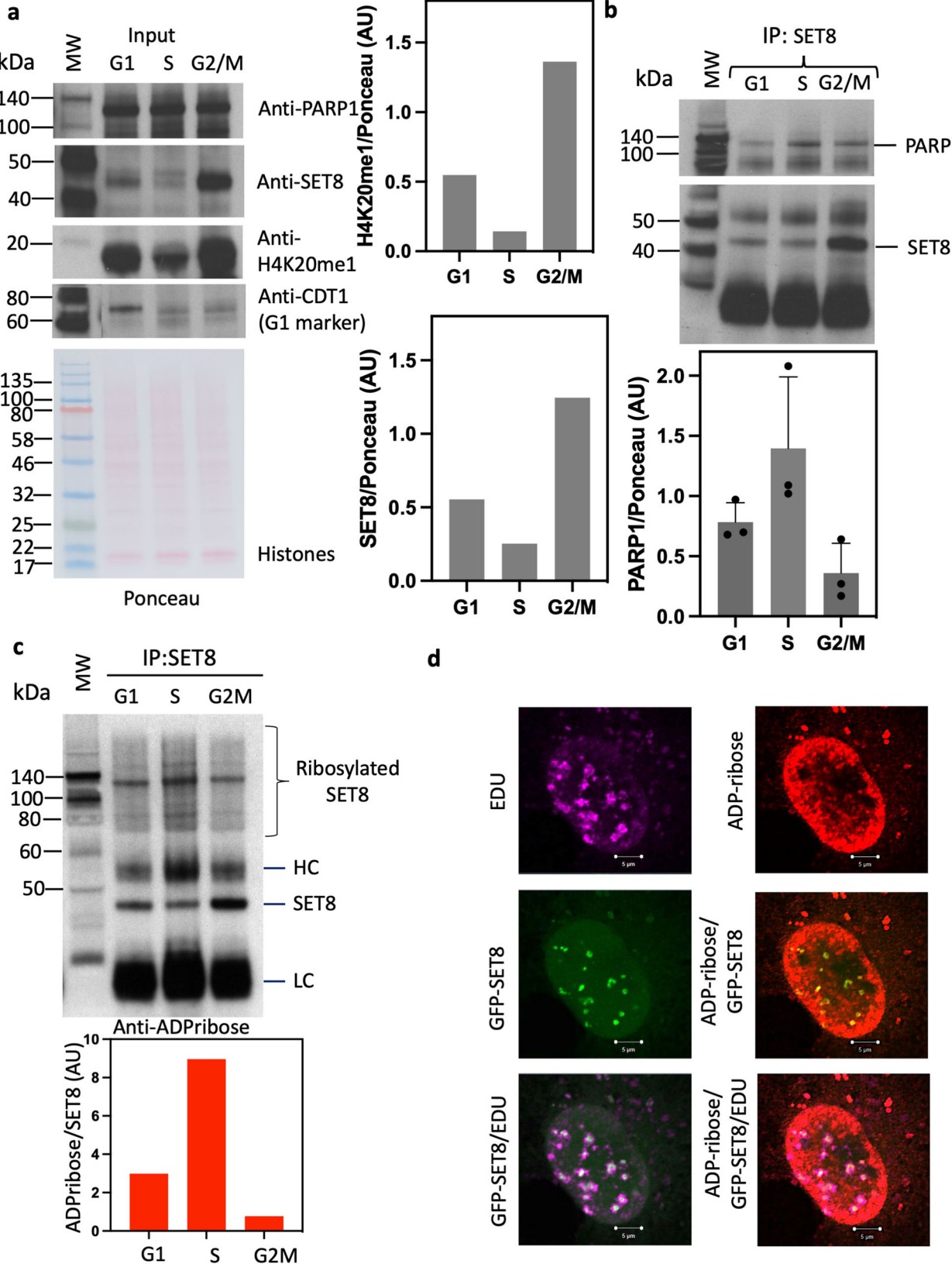

RNAseq also reflects the impact of the H4K20me1 on active transcription (Supplementary Fig. 20a) which is proportionate with the amount of H4K20me1. This was also demonstrated on housekeeping genes, chromatin openness, etc.[55]. Furthermore, SET8-mediated H4K20me1 regulates Pol II promoter-proximal pausing by regulating H4K16ac and H4K20me3 levels and

ultimately transcriptional output[56]. Therefore, PARP1-mediated poly ADP-ribosylation can impact epigenome inheritance, chromatin structure, and transcription factor occupancy.

In summary, our study reveals a novel mechanism of SET8-mediated epigenome regulation. Poly ADP-ribosylation not only can catalytically impair the enzyme activity, but it also triggers a

**Fig. 5 Cell cycle-dependent interaction of PARP1 and SET8 correlates with global H4K20me1. a** Western blot indicating PARP1 (top, left), SET8 (middle, left), and H4K20me1 (middle, left) levels in total protein extracts from HeLa cells synchronized in G1, S, and G2/M phases, respectively. Western blot of CDT1 protein levels is shown as a cell cycle synchronization control as well as Ponceau stain for loading control and densitometry analyses (bottom, left). Respective densitometry analyses of H4K20me1 (top, right; $n = 2$) and SET8 (bottom, right; $n = 2$) relative protein abundances are shown (right) and representative of at least 2 biological experiments. **b** SET8 immunoprecipitation from total protein extract in HeLa cells synchronized in G1, S, and G2/M phases. Western blots detection of PARP1 (top, left) as well as SET8 immunoprecipitated protein levels (bottom, left) are revealed. Densitometry analyses of PARP1/SET8 ratio during G1, S, and G2/M cell cycle phases are shown (right; $n = 3$) and are representative of at least 2 biological experiments. **c** SET8 immunoprecipitation from total protein extract in HeLa cells synchronized in G1, S, and G2/M phases. Western blots detection of ADP-ribosylation as well as SET8 protein levels in SET8 immunoprecipitates (top panel). Respective densitometry analyses of SET8 ADP-ribosylation abundance representative of at least 2 biological experiments are shown (bottom panel; $n = 2$). **d** Pulsed chased cells with 5-ethynyl-2′-deoxyuridine (EdU) to label DNA (magenta) is transfected with GFP-SET8 (green). Endogenous ADP-ribose (red) is revealed by anti-ADP-ribose conjugated with Texas Red. Merged images demonstrate the colocalization of EDU, SET8 and ADP-ribose.

series of events that allows ubiquitination of SET8 molecules in the cells leading to its final degradation. It remains to be seen if poly ADP-ribosylation acts as an allosteric activator for ubiquitin ligases of SET8[57]. Since downregulation of SET8 inhibit progression of hepatocellular carcinoma, this insight may aid in designing SET8 inhibitors that may be useful for cancer treatment[58]. Furthermore, data presented herein may provide insight into studies of PAR-dependent epigenome regulation.

## Methods

**Cell culture, transfections, and cell cycle synchronization**. HeLa, HCT116, and COS-7 cells (All obtained from ATCC) were cultured in DMEM supplemented with 10% fetal bovine serum according to ATCC's recommendations. For PARP1 overexpression, HeLa cells were transfected with 500 ng of 3xFLAG or 3xFLAG-PARP1 plasmids using Fugene HD transfection reagent (Promega, # E2311) according to manufacturer's recommendations. Cells were harvested after 48 h. For PARP1 and SET8 knockdown, HeLa cells were transfected with 10 nM of esiRNA targeting either PARP1 (Millipore-Sigma # EHU050101) or SET8 (Millipore-Sigma # EHU111111) using HiPerfect reagent according to manufacturer's protocol (Qiagen # 301704). 10 nM of esiRNA against EGFP (Millipore-Sigma # EHUEGFP) was used as control and cells were harvested after 48 h.

For cell cycle studies, HeLa cells were synchronized using 2 mM thymidine for 24 h, washed once with media, and released for 8 h with 24 μM dCTP (New England Biolabs # N0441S). 2 mM of thymidine or 0.1 mg/ml of Nocodazole (Millipore-Sigma # SML1665) were added for 14–15 h to arrest the cells in S or G2/M phase, respectively. Cells were synchronized in G1 phase using 20 mM of Lovastatin (Tocris # 1530) for 24 h. Chromatin was extracted using CSK buffer (10 mM Pipes pH 6.8, 300 mM sucrose, 100 mM NaCl, 1.5 mM MgCl₂, and 0.5% Triton X-100 supplemented with protease cocktail inhibitors and PMSF) for 30 min on ice. Cells were centrifuged at $2000 \times g$ at 4 °C for 5 min, washed once with CSK buffer and the pellet (chromatin) was resuspended in buffer containing 50 mM Tris.Cl, pH 7.5, 200 mM NaCl, and 1% SDS. Chromatin was then sonicated (10 cycles of 10 pulses) and used either as total chromatin extract or SET8 immunoprecipitation.

**Western blot and densitometry**. Western blots were performed as previously described[59]. Antibodies against SET8 were obtained from Cell Signaling Technology (# 2996), ABCAM (# ab3798), and Santa Cruz Biotechnology (# sc-515433). Anti-PARP1 (# 9532) and anti-ADPRibose antibody (# 83732S) were purchased from Cell Signaling Technology. Anti-H4K20me1 (# MA5-18067), anti-histone H4K20me2 (# 9759), and anti-H4K20me3 (# 5737) were obtained from Thermo Fisher Scientific and Cell Signaling Technology, respectively, and used at 1/1000 dilution. Anti-β-actin (# 4970S) and anti-GFP (# 11814460001) antibodies were obtained from Cell Signaling Technology and Millipore-Sigma, respectively, and used at a 1:5000 dilution. Densitometry was performed using ImageJ software (W. S. Rasband, National Institutes of Health). All densitometry values (arbitrary units) were normalized to either their respective β-actin or Ponceau S staining. ADP-ribose and ubiquitin quantification levels were normalized to immunoprecipitated SET8 protein levels.

**In vivo ADP-ribosylation and ubiquitination assay**. COS-7 cells were co-transfected with a mixture of GFP, GFP-SET8 FL or GFP-SET8, 3xFLAG or 3xFLAG-PARP1, HA-ubiquitin plasmids, and Fugene HD transfection reagent (Promega, # E2311) according to manufacturer's recommendations. After 48 h, transfected COS-7 cells were treated or not with 50 μM MG132 for 2 h and lysed as described previously[60]. Synchronized GFP-SET8 FL transfected cells were treated for 3 h with 10 μM cullin inhibitor (MLN4924, Selleckem # S7109), PARG inhibitor (PDD 00017273, Tocris # 5952) or with DMSO as control. Cell lysates (100–200 μg) were then immunoprecipitated with GFP antibody (Thermo Fisher

Scientific # G10362). Ubiquitination and ADP-ribosylation were detected by Western blot using HA-tag antibody (Cell signaling Technology # 3724S) or anti-ADPribose antibody, respectively.

**Determination of GFP-SET8 and endogenous SET8 proteins half-life**. For GFP-SET8 proteins half-life study, HeLa cells were transfected with 500 ng of plasmids encoding for GFP, GFP-SET8 FL or GFP-SET8 M. After 24 h, with 50 μg/ml of cycloheximide (Sigma-Aldrich # C4859) was added and cells were collected at different time points until 24 h. For endogenous SET8 protein half-life determination, HeLa cells were transfected with esiRNA as described above. Cells were extracted using RIPA buffer. GFP and SET8 protein levels were detected by Western blot. After densitometry analyses, the half-life of endogenous SET8, GFP-SET8 FL, and GFP-SET8 M were calculated using least square method from PRISM 9 where the known equation of Half-life calculation was fitted for endogenous SET8, GFP-SET8 FL, and GFP-SET8 M, respectively.

**PARP1 and SET8 cloning and mutagenesis**. Sequences coding for PARP1 full-length (cDNA obtained from Origen # RC20708), SET8 full-length[61] were cloned into pEGFP-C2 or pGEX5X.1 vector and transformed into NEB 10-beta competent E.Coli (New England Biolabs # C3019H). Mutagenesis to generate PARP1, SET8 domains as well as point mutations were performed using Q5 site-directed mutagenesis kit (New England Biolabs # E0554S). Primers used for cloning and mutagenesis are provided in Supplementary File 1.

**PARP1 and SET8 interaction studies**. For in vitro interaction studies, PARP1 and SET8 GST-fusion and mutant proteins were induced in Escherichia coli ER2566 cells (New England Biolabs # C2566H) using 0.4 mM IPTG overnight at 16 °C. Purification of recombinant fusion proteins from the bacterial lysate were performed as previously described[62]. For GST pull-down assay, GST or the GST fusion proteins was bound to glutathione-Sepharose beads. The assay was performed by pre-incubating the GST or GST fusion protein beads with 100 μg/ml bovine serum albumin (BSA) and the protein to be studied in a binding buffer (1X PBS with 0.1% Triton X-100) with end over end mixing at 4 °C for 1 h. Beads were washed 3 times for 5 min with a binding buffer containing 1% Triton X-100. The beads were mixed with 1× SDS–PAGE sample loading buffer (New England Biolabs # B7703S) and incubated at 98 °C for 5 min. The protein mixtures were separated on a 10% polyacrylamide Tricine gel (Thermo Fisher Scientific # EC6675BOX). Recombinant PARP1 was purchased from Active Motif (# 81037), SET8 was purified from New England Biolabs (# M0428).

Co-immunoprecipitation of endogenous PARP1 and SET8 were performed with 200 μg of total extract from cross-linked HCT116 or HeLa cells with 1% formaldehyde for 10 min using anti-PARP1 antibody (Cell Signaling Technology # 9532), anti-SET8 antibody (Santa Cruz Biotechnology # sc-515433) or 5 μg of rabbit IgG as a control antibody (Santa Cruz Biotechnology # sc-2027). The immunoprecipitations (IP) were performed overnight with end-over-end mixing at 4 °C in TD buffer (50 mM HEPES, pH 7.5, 250 mM NaCl, and 1% Triton X-100). The antibodies were captured using 50 μl of protein G magnetic beads (S1430, New England Biolabs) incubated for 60 min with end over end mixing at 4 °C. After 3 washes with TD buffer, IP reactions were blotted using anti-PARP1 antibody (Millipore-Sigma # HPA045168) and rabbit anti-SET8 antibody (Cell Signaling Technology # 2996 S). For immunoprecipitation of SET8 from synchronized HeLa cells, 200 μg of chromatin cell lysate was incubated with 5 μg of anti-SET8 (sc-515433, Santa Cruz). IP reactions were performed and blotted with anti-PARP1 and anti-SET8 antibodies as described above.

**Immunofluorescence studies**. For the detection of PARP1 and SET8 colocalization, COS-7 or HeLa cells were grown on coverslips co-transfected with FLAG-PARP1 and GFP-SET8 plasmids as described above. After 12 h, cells were synchronized as described above, cross-linked with 4% paraformaldehyde (Electron Microscopy Sciences # 15710) for 10 min at RT and quenched with 0.125 M Glycine for 5 min at RT. After 20 min permeabilization with 100% methanol for

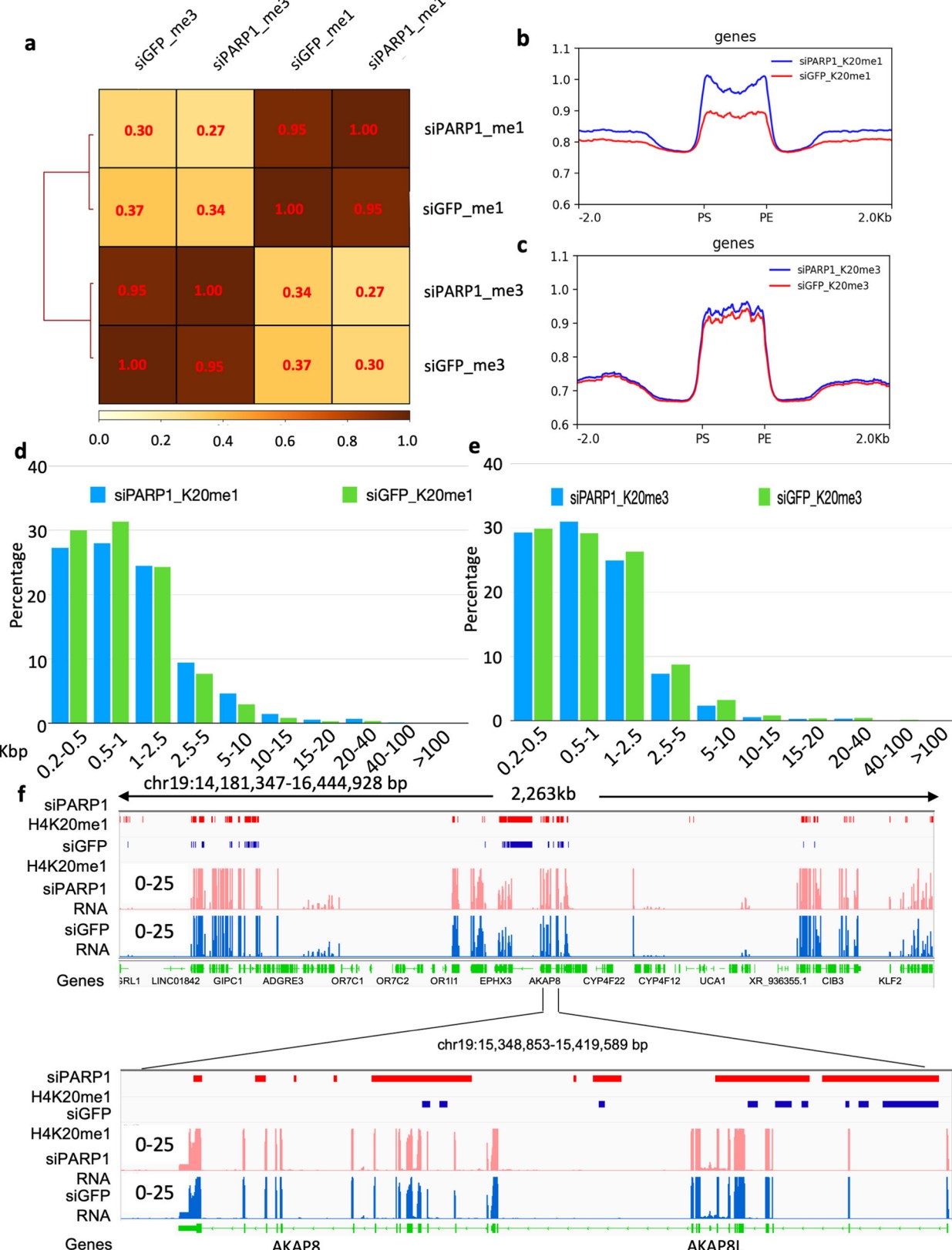

**Fig. 6 PARP1 regulates H4K20me1 and H4K20me3 distribution in the genome. a** Spearman correlation of H4K20me1 and H4K20me3 regions in PARP1 knockdown HeLa cells and its control. **b** Genome-wide metagene plot showing H4K20me1 profile (ChIP-seq) in control and PARP1 knockdown HeLa cells. **c** Genome-wide metagene plot showing H4K20me3 profile (ChIP-seq) in control and PARP1 knockdown HeLa cells. **d** Peak width profile of H4K20me1 by binning the peaks into different lengths in PARP1 knockdown cells and its control. **e** Peak width profile of H4K20me3 by binning the peaks into different lengths in PARP1 knockdown cells and its control. **f** Representative IGV genomic tracks showing H4K20me1 and RNAseq profile in PARP1 knockdown cells and its control.

20 min at −20 °C, cells were incubated with PBS including 0.5% Tween20 and 5% BSA (Millipore-Sigma # A-7906) for 1 h at RT. Epitope-tagged PARP1 was detected by mouse anti-FLAG antibody (# F3165, Millipore-Sigma) and visualized with an anti-mouse IgG coupled with Alexa Fluor 594 dye (Thermo Fisher Scientific # A-11005). GFP-SET8 and FLAG-PARP1 were detected using 458, 488, 514 nm multiline Argon laser and 561 nm DPSS laser, respectively. Endogenous PCNA was visualized with anti-PCNA Alexa Fluor 647 conjugate (Cell Signaling Technology #82968S) using 633 nm HeNe laser. Slides were mounted using Prolong Gold Antifade Reagent with DAPI (Thermo Fisher Scientific # P36931). Images were captured using a confocal microscope (LSM 880, Zeiss).

DNA replication sites were detected using EDU Staining Proliferation Kit (iFluor 647, ABCAM # ab222421) on GFP-SET8 full-length transfected COS-7 cells according to manufacturer's recommendations. Before EDU labeling, GFP-SET8 transfected COS-7 cells were pretreated with 20 μM of PARG inhibitor (Millipore-Sigma # SML1781) for 1 h. After EDU labeling, cells were incubated overnight at 4 °C with anti-pan-ADPribose reagent at 1/400 dilution (Millipore-Sigma # MABE1016). ADP-ribose was detected using anti-rabbit IgG coupled with Alexa Fluor 594 dye (Thermo Fisher Scientific # A32740). EDU was visualized using 633 nm HeNe laser. Slides were mounted and images were captured as described above.

**SET8 gel shift DNA-nucleosome binding assays**. For SET8 DNA gel shift assay, Cy3-end-labeled EcoRI hairpin oligonucleotide (0.5 mM) (GGGAATTCC-CAAAGGGAATTCCC, EcoRI sites underlined) and different concentration of recombinant His-SET8 protein (0 to 1.4 mM, New England Biolabs) were incubated for 10 min on ice in 1× GRB binding buffer [20 mM HEPES pH 7.5, 10% glycerol, 25 mM KCl, 0.1 mM EDTA, 1 mM dithiothreitol (DTT), 2 mM MgCl₂, 0.2% NP40] in a 20 μl reaction volume. Protein–DNA complexes were resolved on a 6% TBE DNA retardation gel (Thermo Fisher Scientific # EC6365BOX) at 4 °C in 0.5× TBE (45 mM Tris-HCl, 45 mM H₃BO₃, 10 mM EDTA pH 8.3) at 140 V. Complex was visualized using Typhoon scanner. Kd was calculated using GraphPad Prism 8.0.

For GST-SET8 full-length as well as mutant DNA or nucleosome binding, 5 μg of GST-beads were incubated for 15 min on ice in GRB buffer in a 25 ml reaction volume with 1 μg of 100 bp DNA ladder (New England Biolabs # N3231S) or mononucleosome (Active motif # 81070). After 1000 rpm spin for 5 min at 4 °C, supernatant representing the unbound DNA was mixed with 6x purple dye (New England Biolabs # B7024S) and loaded on a 6% TBE DNA retardation gel (Thermo Fisher Scientific # EC6365BOX) at 4 °C in 0.5× TBE at 140 V. DNA was visualized under UV Transilluminator. In some cases, GST or GST-SET8 full-length fusion proteins were ADP-ribosylated using PARP1 recombinant enzyme (Active Motif # 81037) during 15 min at RT. The GST beads were then washed twice with PBS + 1% Triton X-100 and 1 M NaCl to remove PARP1 bound to the beads. After 2 other washes with 1X PBS containing 0.1% Triton X-100, beads were resuspended in GRB buffer and incubated with 1 μg of 100 bp DNA ladder (New England Biolabs # N3231S) or mononucleosome (Active motif # 81070) for 10 min on ice. After 1000 rpm spin for 5 min at 4 °C, supernatant representing the unbound DNA was loaded on a 6% TBE DNA retardation gel and detected under UV Transilluminator.

**ADP-ribosylation and SET8 methyltransferase assays**. For ADP-ribosylation assay, PARP1 recombinant enzyme (from 20 to 50 nM per reaction) was incubated with different substrates including either His-SET8 full-length recombinant protein, SET8 peptides or GST, GST-SET8 full-length or mutants and activated using EcoRI hairpin oligo as described above. The reaction was performed in 1x buffer (50 mM Tris.Cl, pH.8, 4 mM MgCl₂, and 250 μM DTT) in the presence or absence of cofactor ß-Nicotinamide adenine dinucleotide (0.5 mM NAD+ per reaction, New England Biolabs # B9007S) at RT during 15 min in 20 ml total reaction volume. The reaction was then loaded on 10% Tricine gels and subjected to Western blotting. ADP-ribosylation was detected by mono/poly-ADP-ribose (Cell Signaling Technology # 83732S) or polyADP-ribose (monoclonal 10H, Santa Cruz Biotechnology # sc-56198) antibodies. For mass spectrometry analyses, 500 nM of PARP1 recombinant enzyme was incubated with 160 μM SET8 peptide (158–170 aa: KKPIKGKQAPRKK and 86–98 aa: KPLAGIYRKREEK) from 1 h to overnight at RT.

Histone methyltransferase assays were carried out as described previously[63]. SET8 recombinant enzyme (New England Biolabs # M0428S) was incubated with recombinant active PARP1 (Trevigen # 4668-100-01) with or without EcoRI hairpin oligo or NAD+ in histone methyltransferase buffer and 6 μM of radiolabeled [3H] AdoMet (Perkin Elmer Life Science # NET155V001MC). Recombinant human histone H4 (New England Biolabs # M2504S) was used as a substrate. Filter disc method was used to process the samples and the [³H]CH3 incorporated into the H4 protein was determined using a liquid scintillation counter.

**LC-MS analysis of poly ADP-ribosylated peptides**. Peptide solutions were analyzed with ProxeonII nLC – LTQ Orbitrap XL by direct injection on Reprosil-Pur C18-AQ 3 μm 25 cm column and eluted at 300 nL/min. Full scan MS was acquired FT Resolution 60k in orbitrap MS. The most abundant three ions were

selected for data-dependent CID MS/MS fragmentation with normalized collision energy 26. Using the same nanoLC conditions, HCD MS/MS fragmentation was acquired using quadrupole-orbitrap with NCE 27. Isotope peak intensity areas were determined with SIEVE 2.2.58 (Thermo Fisher Scientific) following chromatographic alignment. Chromatographic extracted ions were plotted with the manufacturer's software XCalibur. Potential location of ADPr was screened using a match score modeling of fragment ions in CID MS/MS spectra. The matching score was calculated with Peptide Sequence Fragmentation Modeler, Molecular Weight calculator (https://omics.pnl.gov/software/molecular-weight-calculator). The algorithm is based on Sequest Sp preliminary score but differs in treatment of immonium ions[64]. The HCD spectra were mapped manually to the sequence of ADP-ribosylated peptides.

**Chromatin immunoprecipitation sequencing (ChIP-seq)**. HeLa cells knockdown with PARP1 or GFP esiRNA as mentioned above were subjected to ChIP-seq. Briefly, chromatin was extracted as described above and cross-linked with 1% formaldehyde for 10 min and quenched with 0.125 M glycine. After sonication, H4K20me1 IP was performed overnight in TD buffer as described above using 5 μg of H4K20me1 antibody (Thermo Fisher Scientific # MA5-18067). After antibody capture with protein G magnetic beads and 3 washes with TD buffer, beads were incubated at 65 °C overnight in buffer including 50 mM Tris-HCl, pH 7.5, 200 mM NaCl, 1% SDS, and 1.6 U/μl of proteinase K (New England Biolabs # P8107S) to reverse crosslinks. DNA from supernatants was extracted using phenol/chloroform procedure. Between 1 and 10 ng of DNA were used to generate DNA libraries for subsequent sequencing analyses. DNA libraries were made using NEBNext Ultra II DNA Library Prep Kit for Illumina (New England Biolabs # E7645S) according to the manufacturer's recommendations.

**RNA-sequencing (RNA-seq)**. RNA from HeLa cells knockdown PARP1 with esiRNA as mentioned above was extracted using Quick-RNA Miniprep Kit (Zymo Research # R1054). 1 μg of RNA was used to isolate Poly(A) mRNA using NEB-Next Poly(A) mRNA Magnetic Isolation Module (New England Biolabs # E7490S). NEBNext Ultra II directional RNA Library Prep Kit for Illumina (New England Biolabs # E7760S) was used according to the manufacturer's recommendations to generate cDNA and DNA libraries for subsequent DNA sequencing analyses.

**ChIP-seq data processing**. The raw fastq sequences were trimmed using Trim Galore (http://www.bioinformatics.babraham.ac.uk/projects/trim_galore/) to remove the adapters and low-quality sequences. Trimmed reads were mapped to the human reference assembly hg38 using Bowtie2[65]. Aligned reads in bam format were filtered for duplicates and low-quality alignments using Picard (http://broadinstitute.github.io/picard/) and samtools[66]. The aligned bam files of technical replicates were merged using Sambamba[67]. H4K20me1 enriched regions were identified by calling broad peaks (target over input) using MACS2[68] where the parameter broad-cutoff was set to 0.025 for more robustness. Signal tracks were generated using deeptools bamCoverage[69] with the parameters, -normalizeUsing RPKM -of bigwig -e. Spearman correlation analysis was performed using deeptools plotCorrelation[69] function. H4K20me1 peaks were annotated using HOMER[70] annotatePeaks.pl. Repeats elements were annotated using HOMER[70]. H4K20me1 profile in the gene regions for PARP1 from Hela cells for both the conditions were computed with the deeptools computeMatrix and plotProfile[69] functions. Peak length was calculated for all the conditions to estimate the gain and loss of H4K20me1 after PARP1 knock down. Genomic regions were visualized using Integrative genomic viewer (IGV)[71].

**ChromHMM**. ChromHMM[72] was used to create epigenomic segmentations for PARP1 knockdown HeLa cells and their respective controls using bam files for ChIP-seq of H4K20me1, H4K20me3 from siPARP1 and siGFP. A 15-state model was trained using 200 bp bins (LearnModel -b 200) and genome version hg38.

**RNA-seq data analysis**. The raw fastq files were trimmed and quality check was performed using Trim Galore and FASTQC (https://www.bioinformatics.babraham.ac.uk/projects/fastqc/), respectively. All the samples were in good quality. Trimmed reads were mapped to the human reference assembly hg38 using STAR[73]. Transcript abundance was estimated from these high-quality mapped reads using htseq count module[74]. DESeq2[75] was used to normalize the count matrix and identification of differentially expressed genes. Genes that were showing logFC ≥ 0.5 at FDR < 0.05 and logFC ≤ 0.5 at FDR < 0.05 were considered upregulated genes and down-regulated, respectively. In this experiment, two sets of RNA seq data were utilized from siGFP and siPARP1 conditions, respectively, where siGFP was considered as control data. Initially, the positively expressed genes from siGFP and siPARP1 conditions were compared with genes of peak annotated files from intragenic regions of H4K20me1 and H4K20me3 Chip seq data. Subsequently, the transcript abundance, estimated from mapped reads of Chip seq were compared with respected RNAseq expression. More elaborately, the upregulated genes, identified from Chip-Seq data from H4K20me1 and H4K20me3 (where siGFP_H4K20me1 and siGFP_H4K20me3 were used as control) were compared with the corresponding expression values from RNA seq Data and vice-versa. To compare, the data distribution, 2D density plotting from ggplot2 from R

had been considered where kernel distribution estimator (KDE) was used to plot the random variables in terms of gene expression from the two different data sources under a polygonal space. These helped to understand the similarity between expression from Chip-Seq and RNA-Seq.

**Model building**. In this experiment, the monomeric sequence of SET8 was modeled using I-TASSER[76] and considered as Wild Type SETD8 (WT SET8). In this case, the focus was on lysine at residue 86, 158, 159, 162, 164, 168, 169, 170, 174. For the mutated structure, all these positions had been replaced with alanine. I-TASSER produced top five energy minimized models in each case using ab initio method. The top models from WT and mutated SET8 were selected for further studies.

**Studying the folding pattern and predicting the structural disorder**. To observe the nature of the DNA binding region, we have utilized two web-based tools namely, PONDR-VLXT[77] and FoldIndex[78]. PONDR-VLXT is an intrinsically disorder region prediction tool where predictive disorder score ≥0.5 of amino acid residues have been considered as intrinsically disordered residues. Rate of disorderedness can be defined as value of predictive disordered score. Similarly, FoldIndex can calculate the index values for each amino acids ranging between 1 to −1 where any residues having index value lower than zero is considered as unfolded residues. Rate of unfoldedness is highly dependent on the index value.

**Structure network analysis of wild type and mutant SET8**. Analysis and prediction of dynamics associated with complex protein systems can be explained and represented using network architecture. In general, a complex system is composed of elements interacting with one another bound together by links like interactions[79]. Weighted edges characterize the strength of interaction. Overlapping modules can, in turn, be dissected from the network (i.e., communities, groups), formed the modules.

Protein network structure is measured using topology of complex 3D architecture commonly known as Gaussian Network Model (GNM). In this approach, a weighted graph G was constructed that represented a 3D structure, $(V, E) \in G$, where $V$ $(V = V1, V2 \dots Vn)$ represented residues as nodes and $E$ $(E = E1, E2, \dots En)$ weighted edges represented pairwise interaction. The internal motions and intrinsic dynamics of proteins dictate the global protein structure and, hence, the function and activity. Normal Mode analysis (NMA) was utilized for predicting the functional motions in SETD8. Elastic Network Model using C-alpha force field was designed through NMA. Subsequently, the cross-correlation study was performed to identify protein segments with similar oscillation, and a matrix was generated using correlation coefficient score. The correlation-based matrix was further used as a full residue adjacency matrix. These networks were split into a highly correlated coarse-grained community cluster network using the Girvan–Newman clustering method (which was highly dependent on edge betweenness)[80]. Betweenness centrality characterized the regions of a protein that show modifications in coupled oscillations derived from mutant as well as the WT protein. Residues having a significant contribution to the intrinsic dynamics of the protein show high centrality value.

**Statistics and reproducibility**. The raw fastq sequences were trimmed using Trim Galore (http://www.bioinformatics.babraham.ac.uk/projects/trim_galore/) to remove the adapters and low-quality sequences. Trimmed reads were mapped to the human reference assembly hg38 using Bowtie2[65]. Aligned reads in bam format was filtered for duplicates and low-quality alignments using Picard (http://broadinstitute.github.io/picard/) and samtools[66]. The aligned bam files of technical replicates were merged using Sambamba[67]. H4K20me1 enriched regions were identified by calling broad peaks (target over input) using MACS2[68] where the parameter broad-cutoff was set to 0.025 for more robustness. Signal tracks were generated using deeptools bamCoverage[69] with the parameters, -normalizeUsing RPKM -of bigwig -e. Spearman correlation analysis was performed using deeptools plotCorrelation[69] function. H4K20me1 peaks were annotated using HOMER[70] annotatePeaks.pl. Repeats elements were annotated using HOMER[70]. H4K20me1 profile in the gene regions for PARP1 from Hela cells for both the conditions were computed with the deeptools computeMatrix and plotProfile[69] functions. Peak length was calculated for all the conditions to estimate the gain and loss of H4K20me1 after PARP1 knock down. Then the enrichment scores were summarized into a data matrix in R[81] and a heatmap was then created using heatmap.2 function to represent condition-specific enrichment of TF binding motifs near the H4K20me1 peaks. Genomic regions were visualized using Integrative genomic viewer (IGV)[71].

We have 6 ChIP-seq (including 2 input sequences for background subtraction) (siPARP1_H4K20me3 (Rep1/2) = 25374497, siPARP1_H4K20me1 (Rep1/2) = 23044722, siGFP_H4K20me3 (Rep1/2) = 35593041, siGFP_H4K20me1 (Rep1/2) = 27798385, siPARP1_Input (Rep1/2), siGFP_Input (Rep1/2)), and 2 RNAseq Data (siPARP1_RNAseq (Rep1/2), siGFP_RNAseq (Rep1/2)). All reads are paired-end reads.

**Reporting summary**. Further information on research design is available in the Nature Portfolio Reporting Summary linked to this article.

## Data availability

ChIP-seq and RNA-seq data performed in this study are available in NCBI Gene Expression Omnibus (GEO) under the accession GSE188744. Uncropped Western Blot images are given in Supplementary Fig. 21. All the source data for graphs and charts are given in Supplementary Data 1–9. PARP1 full-length (cDNA obtained from Origen # RC20708) and cDNA for SET8 and related clones may be requested from Boston University, ref. [61] (Ula Hansen email-uhansen@bu.edu). Mass spectrometry data set for proteomics analysis of SET8 pull-down in HEK293T cells is available in Supplementary Table 1.

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

## Acknowledgements

We thank C. Carlow for critical reading of the manuscript, T. Evans, D. Comb, Sir R.J. Roberts, J. V. Ellard, and S. Russello for encouragement. The project was funded by basic research grant to SP from the New England Biolabs, Inc.

## Author contributions

P.O.E., C.R., and H.G.C. performed experiments. U.S.V. and S.S. performed data analysis. S.P. supervised this work, conceptualized the project, and wrote the manuscript in collaboration with P.O.E., C.R., H.G.C., U.S.V., and S.S.

## Competing interests

The authors declare no competing interests.
