## [Peer Review File · Communications Biology]

Reviewers' comments:

Reviewer #1 (Remarks to the Author):

In this paper the authors present a functional crosstalk between SET8 and PARP1. They provide evidence using various experimental approaches that SET8 poly ADP-ribosylation by PARP1 leads to aberrant H4K20 methylation. Mechanistically, they present convincing data that SET8 ribosylation leads to its proteasomal degradation. They also provide some data that this occurs in a cell-cycle dependent manner. Overall, it's a well written paper and the take home message is clear. There are some additional controls and clarification that would improve the manuscript. These are summarized below:

- Figure 1A- It will be informative to show a full rather than a cropped image of the input as it seems that relative to the IP which looks very clean, there are many cross reacting bands in the input.
- Figure 1B- the idea of looking at the cross talk between the two proteins during cell cycle is very good. A more informative way to look at it as to use synchronized cells. this will also allow the authors to monitor the kinetics of the interaction and test if the co-localization occurs at a specific stage.
- The author nicely show evidence of ADP-ribosylation using masspec for short synthetic peptide. I was wondering if it is possible to detect endogenous ADP-ribosylation in Masspec.
- The quality of the gel shift assays presented in Figure 2 is low which makes it hard to evaluate the conclusions written in the text. Maybe, lower expression or any kind of quantitation will make the results clearer.
- In Figure S13- a total H4 WB is required to show equal loading. Giving the relative high exposure time, it seems that will there is indeed a significant decrease in K20me1, it seems that me2 and me3 are also going down in a PARP1 dependant manner.
- In Figure 3B the authors decided to use specific inhibitors for Cul4A and PARG and didn't see any changes in the expression of over-expressed SET8. A positive control for these inhibitors could be very informative. SET8 ubiquitination by Cul4A takes place during S phase. I think that these experiments should have been performed in synchronized cell. In addition, the author claims that while there is no change in SET8 level, GFP-SET8 showed more prominent high molecular weight poly ADP-ribosylated in PARGi or Cul4Ai -treated cells. I can see it for PARGi but not for Cul4A1.
- A nice control will be to perform the CHX experiments in the PARP1 KD cells.
- The quality of Fig 6F should be improved. I believe that the authors have described the results correctly in the text but I couldn't observe their conclusions in the Figure. A more "user friendly" figure should be prepared. Maybe increasing the font size in some of the places and highlighting the main conclusion by framing the exact genomic loci they are referring to.
- In the last section of the paper the authors performed a bioinformatics analysis and nicely show an enrichment of consensus TF binding motifs near the K20me1/me3 marks in a PARP1 dependent manner. I think it may be outside the scope of the current paper but it could be nice to validate these observations in direct-chip experiments as it can provide valuable mechanistic angel for their findings.

Reviewer #2 (Remarks to the Author):

In this study, the authors showed that SET8 was post-translationally poly ADP-ribosylated by PARP1 on several lysine residues, leading to SET8 degradation via ubiquitylation and aberrant H4K20me1 and H4K20me3 genome-wide. A lot of work was done in this study, and the manuscript is well-written. The study was very focused. I have just a few comments related to clarification of some points:

- 1.As showed in Figure 1E, the authors did MS analysis, and found multiple lysine residues were poly ADP-ribosylated, including K86, K153, K162 and K164. Then the authors generated SET8 mutants(SET8 M) for further poly ADP-ribosylation study and other functional analysis. Why K153 was not mutated in the SET8 mutants(SET8 M)?
- 2.The quality of some figures needs to be improved:

Figure 3C: SET8 western blot figure.
Figure 4C: Ponceau Stain figure.
Figure 6C

Below is or point by point response to reviewer comments. Changes made in the manuscript is highlighted in red.

Reviewer #1

In this paper the authors present a functional crosstalk between SET8 and PARP1. They provide evidence using various experimental approaches that SET8 poly ADP-ribosylation by PARP1 leads to aberrant H4K20 methylation. Mechanistically, they present convincing data that SET8 ribosylation leads to its proteasomal degradation. They also provide some data that this occurs in a cell-cycle dependent manner. Overall, it's a well written paper and the take home message is clear. There are some additional controls and clarification that would improve the manuscript. These are summarized below:

1.- Figure 1A- It will be informative to show a full rather than a cropped image of the input as it seems that relative to the IP which looks very clean, there are many cross-reacting bands in the input.

Thank you for the detailed critique. Please find the full gel picture. The gel was cut into two halves to keep the IP protein amounts relevant to each other. The upper half was probed with anti-PARP1, and the lower half was probed with Anti-SET8. Non-specific (NS) bands are now shown. The input image of PARP1 has smear due to its poly ADP-ribosylated forms.

2.- Figure 1B- the idea of looking at the cross talk between the two proteins during cell cycle is very good. A more informative way to look at it as to use synchronized cells. this will also allow the authors to monitor the kinetics of the interaction and test if the co-localization occurs at a specific stage.

Please find the cell cycle-based localization studies of PARP1 and SET8 in synchronized cells (Fig 1B and Fig. S1). We calculated Pearson's correlation between both proteins to monitor kinetics. The text is modified as "For this experiment, we transfected COS-7 cells with FLAG-PARP1 and GFP-SET8, synchronized the cells and studied their association using confocal microscopy and Pearson's correlation coefficient during G1, S and G2/M stages. At G1, GFP-SET8 and FLAG-PARP1 remain distributed throughout the nucleus, compared to S phase where punctate pattern of both GFP-SET8 and FLAG-PARP1 was observed. However, as expected FLAG-PARP1 remained throughout the nucleus, and appeared as prominent punctate foci with GFP-SET8, as observed by bright yellow merged spots (Fig 1B). This was further supported by their association kinetics during cell cycle where the Pearson's correlation was higher during S phase ($r=0.6$) compared to G1 ($r=0.3$) and no correlation was observed in G2/M phase (Fig. S1A)".

3.- The author nicely show evidence of ADP-ribosylation using masspec for short synthetic peptide. I was wondering if it is possible to detect endogenous ADP-ribosylation in Masspec.

Poly ADP-ribosylation is highly heterogenous between protein molecule to molecule. Ribosylation also adds charge to the protein. Our effort to perform endogenous poly ADP-

ribosylation using Mass spectroscopy was unsuccessful. Therefore, we performed pain-striking deletion analysis to prove and identify those sites.

4. The quality of the gel shift assays presented in Figure 2 is low which makes it hard to evaluate the conclusions written in the text. Maybe, lower expression or any kind of quantitation will make the results clearer.

Please find lower expression picture with quantitation in Fig S13. The text is modified as “This binding was partially dependent on the 157-352 amino acids of SET8, and a significant loss of binding was observed in a deletion mutant comprising 175-352 amino acids, suggesting 157-175 amino acids play a functional role in DNA binding (Fig. 2A, Fig. S13A).”

One more sentence was also added as “Once again, 157-175 amino acids played a functional role in nucleosome binding (Fig. 2C, Fig. S13B).”

5. In Figure S13- a total H4 WB is required to show equal loading. Giving the relative high exposure time, it seems that there is indeed a significant decrease in K20me1, it seems that me2 and me3 are also going down in a PARP1 dependent manner.

Please find the H4 WB picture included in Fig S16.

6. In Figure 3B the authors decided to use specific inhibitors for Cul4A and PARG and didn't see any changes in the expression of over-expressed SET8. A positive control for these inhibitors could be very informative. SET8 ubiquitination by Cul4A takes place during S phase. I think that these experiments should have been performed in synchronized cell. In addition, the author claims that while there is no change in SET8 level, GFP-SET8 showed more prominent high molecular weight poly ADP-ribosylated in PARGi or Cul4Ai -treated cells. I can see it for PARGi but not for Cul4A1.

In figure 3B the densitometric scan and quantification is now included on top of the bottom two blots. Control DMSO/GFP-SET8 is taken as 1X and Cul4Ai and PARGi are indicated as relative to control. Detection and quantification of fusion GFP-SET8 Ubiquitination were added in Fig 3B 4th panel. Text was modified “When these samples were western blotted and probed with anti-HA antibody to decipher ubiquitin abundance, we observed almost 40% reduction in ubiquitination confirming Cul4A inhibitor (MLN4924) essentially act by inhibiting ubiquitination pathway. Similarly, upon PARGi treatment 1.6X GFP-SET8 accumulation is visible by high molecular weight poly-ubiquitinated fusion enzyme (Fig. 3B, 4th panel).”

7. A nice control will be to perform the CHX experiments in the PARP1 KD cells.

Please find a new Fig. S17 demonstrating the PARP1 knock down experiment. We have modified the text as “We next pursued half-life studies of endogenous SET8 by PARP1 depletion using siPARP1. Half-life of endogenous SET8 in control cells with siGFP transfection was 0.74 hrs

compared to 1.47 hrs in PARP1 knockdown cells (Fig. S17A-B). The half-life of endogenous SET8 of 0.74 hrs matched with previously reported study (19). The discrepancy of half-life between GFP-SET8 (3.8 hrs) vs. endogenous SET8 (0.74 hrs) could be due to the stability of GFP fusion partner of SET8.”

8. The quality of Fig 6F should be improved. I believe that the authors have described the results correctly in the text but I couldn't observe their conclusions in the Figure. A more “user friendly” figure should be prepared. Maybe increasing the font size in some of the places and highlighting the main conclusion by framing the exact genomic loci they are referring to.
- In the last section of the paper the authors performed a bioinformatics analysis and nicely show an enrichment of consensus TF binding motifs near the K20me1/me3 marks in a PARP1 dependent manner. I think it may be outside the scope of the current paper but it could be nice to validate these observations in direct-chip experiments as it can provide valuable mechanistic insight for their findings.

Please find a new Fig. 6 with larger font size and indicating the genome loci. The text reflects these changes “IGV browser displayed changes in H4K20me1 boundaries throughout the genome upon PARP1 knockdown (red segments) compared to control siGFP (blue segments) knockdown as observed for AKAP8 (Fig. 6F).”

Reviewer #2 (Remarks to the Author):

In this study, the authors showed that SET8 was post-translationally poly ADP-ribosylated by PARP1 on several lysine residues, leading to SET8 degradation via ubiquitylation and aberrant H4K20me1 and H4K20me3 genome-wide. A lot of work was done in this study, and the manuscript is well-written. The study was very focused. I have just a few comments related to clarification of some points:

1. As shown in Figure 1E, the authors did MS analysis, and found multiple lysine residues were poly ADP-ribosylated, including K86, K153, K162 and K164. Then the authors generated SET8 mutants (SET8 M) for further poly ADP-ribosylation study and other functional analysis. Why K153 was not mutated in the SET8 mutants (SET8 M)?

Thank you for pointing this error from our end. It is indeed K158. We have modified the manuscript throughout.

2. The quality of some figures needs to be improved:
Figure 3C: SET8 western blot figure.

Please find a new figure indicating SET8 (bottom panel).

Figure 4C: Ponceau Stain figure.

Please find improved figure 4C.

Figure 6C

Please find improved figure 4C with clear figure with bigger fonts and highlights.

REVIEWERS' COMMENTS:

Reviewer #1 (Remarks to the Author):

The authors have nicely addressed the main concerns I had and I suggest accepting it for publication.

Below is or point by point response to reviewer comments. Changes made in the manuscript is highlighted in red.

Reviewer #1

The authors have nicely addressed the main concerns I had and I suggest accepting it for publication.

Thank you for accepting it for publication.